# Turbulence simultaneously stimulates small- and large-scale $CO_2$ sequestration by chain-forming diatoms in the sea

Johanna Bergkvist[1], Isabell Klawonn[2,7], Martin J. Whitehouse [3], Gaute Lavik[4], Volker Brüchert[5] & Helle Ploug[1,6]

Chain-forming diatoms are key $CO_2$-fixing organisms in the ocean. Under turbulent conditions they form fast-sinking aggregates that are exported from the upper sunlit ocean to the ocean interior. A decade-old paradigm states that primary production in chain-forming diatoms is stimulated by turbulence. Yet, direct measurements of cell-specific primary production in individual field populations of chain-forming diatoms are poorly documented. Here we measured cell-specific carbon, nitrate and ammonium assimilation in two field populations of chain-forming diatoms (*Skeletonema* and *Chaetoceros*) at low-nutrient concentrations under still conditions and turbulent shear using secondary ion mass spectrometry combined with stable isotopic tracers and compared our data with those predicted by mass transfer theory. Turbulent shear significantly increases cell-specific C assimilation compared to still conditions in the cells/chains that also form fast-sinking, aggregates rich in carbon and ammonium. Thus, turbulence simultaneously stimulates small-scale biological $CO_2$ assimilation and large-scale biogeochemical C and N cycles in the ocean.

Diatoms contribute ca. 20% to the total primary production on Earth and play a key role in $CO_2$ sequestration in the ocean[1]. Chain-forming genera of diatoms, e.g. *Chaetoceros, Skeletonema* and *Thalassiosira*, are dominant primary producers in environments with high turbulent shear, silicate, and nitrate concentrations, e.g., spring blooms in the polar oceans and temperate regions, and in subtropical and tropical upwelling regions. During these blooms, the formation of millimeter-sized, fast-sinking diatom aggregates in the upper sunlit ocean are instrumental for connecting the surface and the deeper ocean with respect to aggregate-attached biota, carbon (C) and nutrient sources, the aerobic and anaerobic nitrogen (N) cycles, and for biological $CO_2$ sequestration to the mesopelagic ocean[2–7].

Over the past half century, several theories have been developed to explain the global success of chain-forming diatoms, and how chain formation may provide adaptations to physical, chemical

[1] Department of Biological and Environmental Sciences, University of Gothenburg, Box 461SE-405 30 Gothenburg, Sweden. [2] Department of Ecology, Environment and Plant Sciences, Stockholm University, Svante Arrhenius Väg 21A, SE-10691 Stockholm, Sweden. [3] Swedish Museum of Natural History, Department of Geological Sciences, Stockholm University, SE-10405 Stockholm, Sweden. [4] Max Planck Institute for Marine Microbiology, Celsiusstr. 1, D-28359 Bremen, Germany. [5] Department of Geological Sciences, Svante Arrhenius Väg 8, Stockholm University, SE-10691 Stockholm, Sweden. [6] Department of Marine Sciences, University of Gothenburg, Box 460SE-405 30 Gothenburg, Sweden. [7] Present address: Leibniz Institute of Freshwater Ecology and Inland Fisheries (IGB), Stechlin 16775, Germany. These authors contributed equally: Johanna Bergkvist, Isabell Klawonn. Correspondence and requests for materials should be addressed to H.P. (email: Helle.Ploug@marine.gu.se)

and biological constraints in the sea[8–17]. Laboratory experiments with diatom cultures have shown that chain formation can facilitate nutrient uptake under turbulent shear[10–13]. However, the role of turbulent shear for cell-specific C-, and nitrate assimilation by chain-forming diatoms within mixed field communities, and the relation between nutrient demand and supply in co-existing individual diatom cells and cell chains remain poorly understood due to technical and methodological limitations.

Mixing of water is inefficient at a micrometer scale where viscous forces dominate inertial forces. Thus, transport of gases and nutrients between phytoplankton cells and ambient water occurs by diffusion through the diffusive boundary layer (DBL) in the water adjacent to the cells and cell chains (Fig. 1a). Diffusion-limited primary production occurs when cellular demand and potential uptake rate of gases or nutrients are higher than the maximum diffusive supply from the ambient water to the cell, which is constrained by the concentration gradients within the DBL at the cell–water interface (Eq. (1)). Small phytoplankton cells or cell colonies can theoretically cover their C and nutrient demands faster at low concentrations in ambient water, primarily due to shorter diffusion distances, lower cellular C and nutrient demands, and larger surface area:volume ratios compared to larger phytoplankton cells or colonies. The active nutrient uptake across the cell membrane keeps concentration low at the cell surface relative to that in the ambient water, and a large surface area:volume ratio of small cells allows for more ion carriers across the cell membrane relative to their cell volume compared to large cells. Diffusion-limited primary production occurs when the concentration gradients across the DBL limit the supply of gases or nutrients from the ambient water, and therefore occurs more likely in large cells or colonies of cells than in small, free-living cells when ambient nutrient concentrations are low[18–20].

The Kolmogorov length scale describes the size of the smallest eddies due to turbulence and is usually on the order of 1 to 10 mm in the surface ocean[21]. Below this scale fluid flow is laminar and the turbulent energy is dissipated through viscous shear, which enhances diffusion-limited gas and nutrient exchange in large phytoplankton (cells >60 μm in radius)[8,9]. Turbulent shear is theoretically more significant for mass transfer to large cells compared to small cells (Fig. 1b; Eq. (2)) and may alleviate diffusion-limited fluxes of gases and nutrients in large cells or colonies[9–13,18–22].

Secondary ion mass spectrometry (SIMS) combined with stable isotopic tracers allows insight into life at the single-cell level of micrometer-sized individuals in mixed microbial field populations[23–27]. SIMS combines the qualities of a microscope with those of a mass spectrometer and reveals elemental and isotopic compositions of single cells of known identity and their individual biological activity. Using SIMS, we tested three central hypotheses in biological oceanography and biogeochemistry in two field populations of chain-forming diatoms. The first hypothesis is that turbulence enhances C assimilation and nutrient fluxes in chain-forming diatoms. The second hypothesis is that small cells grow faster than large cells under still or low, turbulent shear at low ambient nutrient concentrations. The third hypothesis is that aggregates of chain-forming diatoms mediate significant ammonium fluxes to nitrogen-depleted, ambient water. To test these hypotheses, we analysed cell-specific C-fluxes and N-fluxes in two co-existing field populations of the chain-forming diatoms *Skeletonema* and *Chaetoceros* using SIMS combined with stable isotopic tracers and compared our data with those predicted by mass transfer theory. By combining sensitive fluorometry and mass transfer theory, we quantified ammonium concentrations within aggregates and fluxes from aggregates to the ambient water.

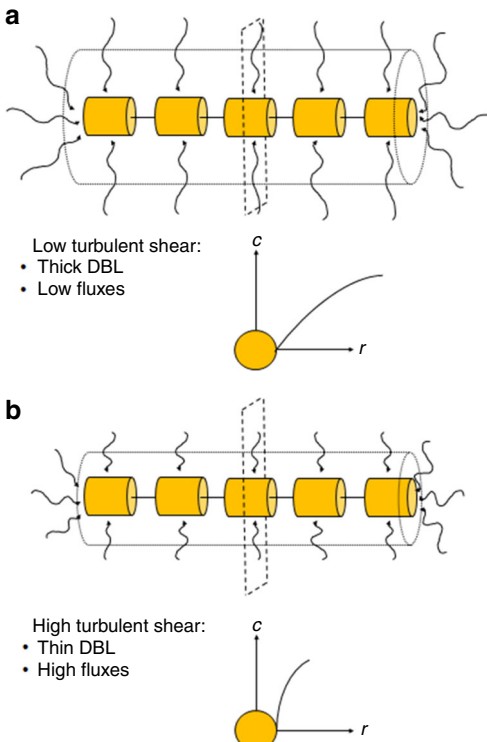

**Fig. 1** Diffusive boundary layers (DBL) surrounding chain-forming diatoms. The radial diffusion-limited concentration gradient across the DBL from the ambient water to cell surface for the middle cell under low turbulent shear (**a**) and under high turbulent shear (**b**)

Our study showed that turbulent shear indeed increases C assimilation in chain-forming diatoms as predicted by mass transfer theory. However, small cells do not always grow faster than large cells when comparing field populations of different diatom genera. Finally, aggregates of chain-forming diatoms do mediate a substantial ammonium flux to nitrogen-depleted water.

## Results

**Phytoplankton community composition**. Experiments were performed with Baltic Sea spring bloom populations during April 2013. The fraction of large cells (>2 μm) in the phytoplankton community was dominated by the diatom genera *Skeletonema* and *Chaetoceros* (Supplementary Fig. 1). Average chains consisted of seven cells for *Skeletonema* (maximum: 22 cells) and four cells for *Chaetoceros* (maximum: 12 cells chain$^{-1}$) (Table 1). Less abundant phytoplankton were the diatoms *Coscinodiscus* sp., *Melosira arctica*, *Thalassiosira* sp. and *Navicula* sp., and the dinoflagellate *Peridiniella catenata*. Picoplankton (<2 μm) was presumably also present, but was not quantified here.

**Community primary production and nitrogen assimilation**. During sampling, the spring bloom was well-developed as reflected by Chl. *a* concentrations, which had a mean value of 13 μg L$^{-1}$. Suspended POC and PON concentrations were 57.8 μmol C L$^{-1}$ and 5.9 μmol N L$^{-1}$, respectively (Table 2), and the concentration of silicate was high (5.6 μM) relative to inorganic nitrogen and phosphate concentrations (<0.3 μM). The POC:PON (mol:mol) ratio of total organic matter and the ratio of C to nitrate plus ammonium assimilation were in the range of 8.5 to 10.6 indicating N-limitation of primary production by the phytoplankton community. Nitrate was the primary source of N-uptake within the mixed community as the nitrate assimilation

| Table 1 Characteristics of *Skeletonema* and *Chaetoceros* | | | | | | | |
|---|---|---|---|---|---|---|---|
| **Genera** | **Cell length (μm)** | **Cell width (μm)** | **Cell volume (μm³)** | **Cellular POC cont.[a] (pmol C cell⁻¹)** | **Cellular PON cont.[a] (pmol N cell⁻¹)** | **Chain length (cells chain⁻¹)** | **Abundance (cells L⁻¹)** |
| *Skeletonema* | 15 | 5 | 294 | 2.4 | 0.34 | 7.0 | $2.023 \times 10^6$ |
| *Chaetoceros* | 10 | 15 | 1766 | 10.3 | 1.6 | 4.3 | $1.486 \times 10^6$ |

Cell dimensions, cellular POC and PON content, chain length and abundance
POC: particulate organic carbon, PON: particulate organic nitrogen
[a]Calculated from the cellular dimensions (Menden-Deuer and Lessard[28])

| Table 2 Chemical parameters and assimilation rates | |
|---|---|
| Nitrite + Nitrate (μmol L⁻¹) | $0.28 \pm 0.36$ |
| Ammonium (μmol L⁻¹) | $0.176 \pm 0.107$ |
| Phosphate (μmol L⁻¹) | $0.067 \pm 0.005$ |
| Silicate (μmol L⁻¹) | $5.58 \pm 0.36$ |
| Chl. *a* (μg L⁻¹) | $13.0 \pm 4.0$ |
| Chl. *a*: POC (w:w) | 0.018 |
| POC (μmol L⁻¹) | $57.8 \pm 5.8$ |
| PON (μmol L⁻¹) | $5.9 \pm 0.5$ |
| C:N ratio (mol:mol) | 9.8 |
| C-assimilation rate (μmol C L⁻¹ h⁻¹) under still conditions[a] | $1.56 \pm 0.34$ |
| C-assimilation rate (μmol C L⁻¹ h⁻¹) under turbulent shear[a] | $1.21 \pm 0.15$ |
| Nitrate-assimilation rate (μmol N L⁻¹ h⁻¹) under still conditions[a] | $0.147 \pm 0.016$ |
| Nitrate-assimilation rate (μmol N L⁻¹ h⁻¹) under turbulent shear[a] | $0.137 \pm 0.039$ |
| Ammonium-assimilation rate (μmol N L⁻¹ h⁻¹) under still conditions | $0.023 \pm 0.001$ |
| C:Nitrate assimilation ratio under still conditions | 10.6 |
| C:Nitrate assimilation ratio under turbulent shear | 8.8 |
| C: (nitrate + ammonium) assimilation ratio under still conditions | 8.5 |
| Ammonium production rate (μmol N L⁻¹ h⁻¹) under still conditions | $0.019 \pm 0.001$ |

Nutrient concentrations, Chl.a, POC, and PON in ambient water. C-, nitrate and ammonium assimilation rates, and ammonium production rates of the phytoplankton community. The numbers are average values ± standard deviation of the mean value ($n = 3$)
POC: particulate organic carbon, PON: particulate organic nitrogen
[a]Values of community C- and nitrate-assimilation rates were statistically similar under still conditions and under turbulent shear (ANOVA two-way, $p > 0.05$)

rate was 6.4-fold higher than the ammonium assimilation rate. C- and nitrate-assimilation rates by the total phytoplankton community were statistically similar under still conditions and under turbulent shear (two-way ANOVA, $p > 0.13$, DF:4).

**Cell-specific C and N assimilation.** C and N assimilation in single cells within individual cell chains were quantified from $^{13}C$:$^{12}C$ and $^{15}N$:$^{14}N$ SIMS images (Fig. 2). The stable C- and N-isotope composition in individual cells revealed that C- and nitrate-assimilation rates in individual cells of both diatom populations varied by more than one order of magnitude independent of still or turbulent conditions (Fig. 3). Under still conditions, the average C:N (mol: mol) assimilation ratio and the standard error of the average value measured in individual cells was $4.5 \pm 0.3$ and $5.6 \pm 0.9$ in *Skeletonema* and *Chaetoceros*, respectively. Under turbulent shear, it was $5.9 \pm 0.7$ and $4.5 \pm 0.4$ in *Skeletonema* and *Chaetoceros*, respectively. Thus, it was slightly below Redfield ratio of 6.6 independent of still conditions or turbulent shear. Thirty-one percent more C was assimilated in *Skeletonema* cells under turbulent shear relative to still conditions. We used two-way ANOVA to test if cell-specific C- and N-assimilation rates in *Chaetoceros* or *Skeletonema*, were significantly different under still and turbulent conditions as proposed by our first hypothesis. In the *Skeletonema* population, we found that C-assimilation was significantly higher under turbulent shear relative to still conditions (two-way ANOVA, $p < 0.0001$, DF: 598), but nitrate assimilation was not significantly increased under turbulent shear compared to still conditions. Turbulent shear caused no statistically significant changes in the cell-specific C- and nitrate-assimilation rates in the overall *Chaetoceros* population (two-way

ANOVA, $p > 0.23$, DF: 598). However, C assimilation in individual cells of long *Chaetoceros* cell chains (>7 cells chain⁻¹) differed significantly between still conditions and turbulent shear (Fig. 4). Under still conditions, the average C assimilation was 45% lower in cells positioned in the middle of a *Chaetoceros* chain compared to that in apical cells. Under turbulent shear C assimilation did not change in apical cells compared to still conditions, but increased by 59% in cells positioned in the middle of the chains. Consequently, under turbulent shear C assimilation was evenly distributed among *Chaetoceros* cells within the same chain. N assimilation showed a higher variability among the cells and was not statistically different between still and turbulent conditions. In contrast, C and nitrate assimilation rates were uniformly distributed within *Skeletonema* cell chains under still and turbulent conditions. The lower C assimilation in *Chaetoceros* cells positioned in the middle of chains relative to that of end cells under still conditions as well as the increase in C assimilation in the middle of chains relative to that of end cells under turbulent shear were statistical significant (two-way ANOVA, $p < 0.05$; DF:57). Thus, turbulent shear increased C assimilation both within *Skeletonema* and *Chaetoceros* cell chains relative to that under still conditions as proposed by the first hypothesis.

**Mass transfer theory.** The calculated, diffusion-limited DIC supply with respect to bicarbonate to *Chaetoceros* was 167,000 fmol C cell⁻¹ h⁻¹ and more than 1000-times higher than the actual net C-assimilation rate of 67 to 120 fmol C cell⁻¹ h⁻¹, depending on the cell position in the chain (Eq. (1); Fig. 4). Under still conditions, the diffusion-limited supply of nitrate from the

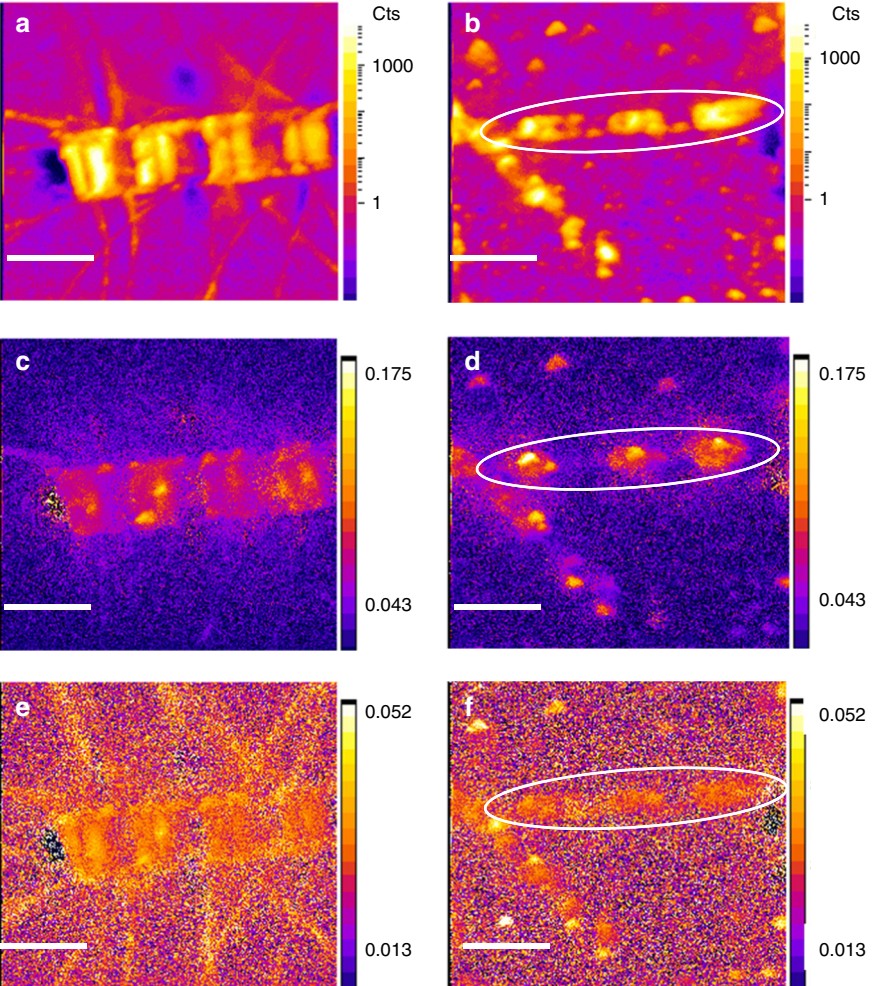

**Fig. 2** SIMS images of *Chaetoceros* and *Skeletonema* cell chains. *Chaetoceros* (**a**, **c**, **e**) and *Skeletonema* (**b**, **d**, **f**) incubated with $^{13}C$-$HCO_3^-$ and $^{15}N$-$NO_3^-$. The biomass distributions are demonstrated by the $^{12}C^{14}N^-$ images (**a**, **b**). The unit represent counts per pixel (**a**, **b**). SIMS images of $^{15}N$:$^{14}N$ ratio (**c**, **d**) and $^{13}C$:$^{12}C$ ratio (**e**, **f**) reflect nitrate assimilation and C assimilation, respectively. The scale bars represent 20 µm

ambient water to *Chaetoceros* cell chains was 38.4 fmol N cell$^{-1}$ h$^{-1}$, which was similar to the average measured nitrate assimilation rates of 37.6 fmol N cell$^{-1}$ h$^{-1}$ (Fig. 4).

The calculated diffusion-limited DIC supply to *Skeletonema* was 114,000 fmol C cell$^{-1}$ h$^{-1}$ (Eq. (1)), but the measured net C assimilation was only 15 to 20 fmol C cell$^{-1}$ h$^{-1}$, i.e. more than 6000 times lower than the potential DIC flux (Fig. 4). Measured nitrate assimilation rates (5.5 fmol C cell$^{-1}$ h$^{-1}$) were 20% of the diffusion-limited supply of nitrate from the ambient water under still conditions (26 fmol N cell$^{-1}$ h$^{-1}$), indicating that primary production and cellular growth in *Skeletonema* was N-limited, but possibly also limited by another element, e.g. phosphorus (P).

Assuming Redfield stoichiometry (N:P assimilation ratio of 16), the P assimilation rate was 2.4 fmol P cell$^{-1}$ h$^{-1}$ and 0.3 fmol P cell$^{-1}$ h$^{-1}$ for *Chaetoceros* and *Skeletonema*, respectively. By comparison, the diffusion-limited P-supply was 3.6 fmol P cell$^{-1}$ h$^{-1}$ and 2.5 fmol P cell$^{-1}$ h$^{-1}$ for *Chaetoceros* and *Skeletonema*, respectively. Hence, the estimated P assimilation rate was 67% of the diffusion-limited P-supply in *Chaetoceros*, while it was 14% of that in *Skeletonema*.

The Sherwood number (Sh) describes the relative increase in total flux of gases or nutrients due to turbulent shear compared to still conditions as predicted by mass transfer theory[9,22]. The predicted increase in total fluxes of nitrate and ortho-phosphate due to turbulence relative to still conditions was 38% and 62%,

respectively, in *Chaetoceros*, whereas they were 25% and 40%, respectively, in *Skeletonema* (cf. Eq. (2)). Our empirical observations of <59% and 31% in *Chaetoceros* and *Skeletonema*, respectively, were thus within the expected theoretical range for N and P coupled to the net C-assimilation rates in the diatoms (Discussion section).

**Cell-specific contributions to community C and N assimilation.** The average biovolume of single *Chaetoceros* cells was six-fold larger than that of *Skeletonema* cells (Table 1), and the average cell-specific C- and N-assimilation rates were sixfold to sevenfold higher in *Chaetoceros* compared to *Skeletonema*. The C-assimilation rate was similar during our ammonium and nitrate assimilation experiments in both *Chaetoceros* and *Skeletonema* under still conditions. In both diatom genera, the cell-specific ammonium assimilation rate was ca. 16% of the nitrate assimilation rate (Fig. 5a). Under still conditions, the *Chaetoceros* population and the *Skeletonema* populations contributed 38% and 8% to the total nitrate assimilation by the phytoplankton community, respectively (Table 3). For ammonium and C assimilation, their relative contributions were 35% and 6%, and 10% and 2%, respectively. Under turbulent shear, their relative contribution to C assimilation was 15% and 4%, respectively, whereas their relative contribution to nitrate assimilation was 32% and 6%, respectively.

The sixfold difference in biovolume between *Chaetoceros* and *Skeletonema* can partly be explained by silicified cell walls as the C and N contents of *Chaetoceros* cells are approximately only 4-fold larger than those of *Skeletonema* cells[28]. C- and N-assimilation rates and doubling times can be expressed independent of cell size when C- and N-assimilation rates are normalised

to cellular C and N contents as measured by SIMS. The average C-specific C-assimilation rate (C-growth rate) in *Chaetoceros* was 40% higher than in *Skeletonema* (cf. Eq. (3); Fig. 5c, d), and the average N-specific N-assimilation rates (N-growth rate) in *Chaetoceros*, with respect to nitrate and ammonium, were 67% and 97% higher, respectively, relative to those in *Skeletonema*. C-growth rates corresponded to average doubling times of 66 and 92 light hours under still conditions, and 62 and 71 light hours under turbulent shear in *Chaetoceros* and *Skeletonema*, respectively. N doubling times were 40 and 66 h under still conditions, and 37 and 56 h under turbulent shear in *Chaetoceros* and *Skeletonema*, respectively. We used two-way ANOVA to test if C and N doubling times were statistically significantly different in *Chaetoceros* vs. *Skeletonema* as proposed by our second hypothesis. Contrary to mass transfer theory, the C and N doubling times were significantly shorter in the larger *Chaetoceros* cells compared to the smaller *Skeletonema* cells both under still conditions and turbulent shear ($p < 0.001$; DF: 598).

**$NH_4^+$ production in diatom aggregates and ambient water**. Fast-sinking diatom aggregates are instrumental in $CO_2$ sequestration in the ocean. Formation of mm-sized diatom aggregates from natural seawater in the laboratory also created ammonium-rich microenvironments that released ammonium to the ambient water (Fig. 6). $NH_4^+$ concentrations within aggregates were in the mid micromolar range, as modeled from the measured $NH_4^+$ release to the ambient water, and were <100-fold higher than in ambient water (176 nM) (Table 1). On average three mm-sized aggregates $L^{-1}$ formed, and the measured average net flux of $NH_4^+$ from individual aggregate to the ambient water was 1.5 nmol $NH_4^+$ $agg^{-1}$ $h^{-1}$, or 4.5 nmol $NH_4^+$ $L^{-1}$ $h^{-1}$. The ammonium production rate measured in these aggregates (nmol N $L^{-1}$ $h^{-1}$) was equivalent to 24% of the ammonium production in the ambient water without aggregates (19 nmol N $L^{-1}$ $h^{-1}$), and equivalent to 6% of the nitrate assimilation rate (147 nmol N $L^{-1}$ $h^{-1}$) by the total phytoplankton community. Thus, diatom aggregates provided a substantial flux of ammonium to the ambient water as proposed in our third hypothesis. In

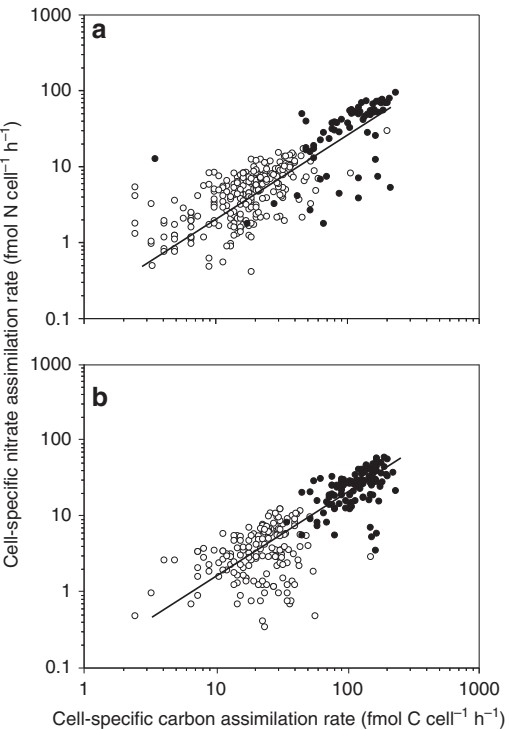

**Fig. 3** C- and nitrate-assimilation rates in individual cells. Each symbol represents the rates measured in individual cells of *Skeletonema* (open symbols) and of *Chaetoceros* (closed symbols) under still conditions (**a**) and under turbulent shear (**b**). The solid lines represent the Redfield ratio

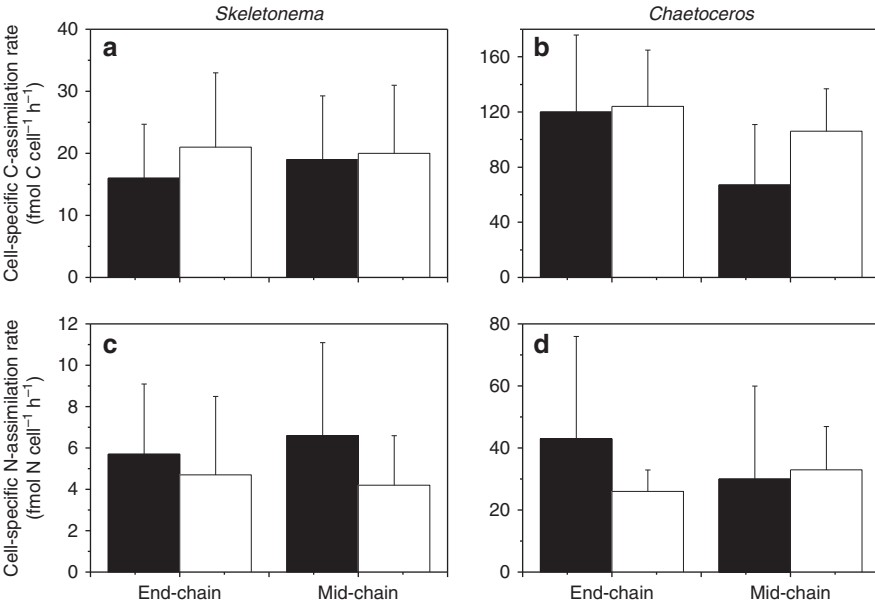

**Fig. 4** C- and N-assimilation rates depending on position within cell chains. Cell-specific C- and nitrate (N) assimilation rate measured in cells positioned at the end or in the middle of cell chains (>7 cells $chain^{-1}$). *Skeletonema* (**a**, **c**) and *Chaetoceros* (**b**, **d**) under still (black bars) and turbulent conditions (open bars). The bars represent the mean value and the error bars one standard deviation ($n = 7$–$28$)

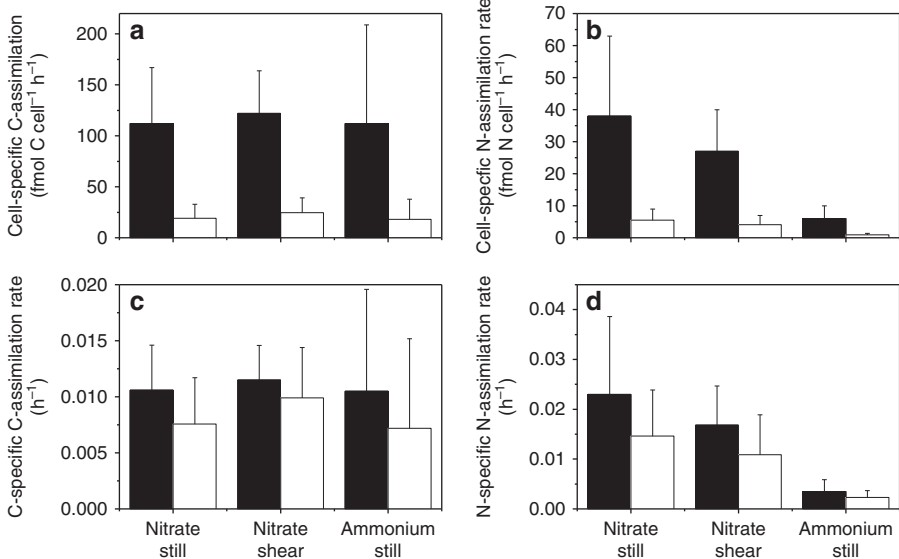

**Fig. 5** C- and N-assimilation rates in the *Skeletonema* population and the *Chaetoceros* population. Cell-specific C-assimilation rates (**a**); Cell-specific N-assimilation rates (**b**); C-specific C-assimilation rates (**c**) and N-specific N-assimilation rates (**d**) in *Chaetoceros* (black bars) and *Skeletonema* (white bars) measured for nitrate as N sources and under still conditions or turbulent shear and at light intensities >200 μmol photons m$^{-2}$ s$^{-1}$. Ammonium assimilation rates were measured without turbulent shear and represent the average value in light and darkness. All values represent the average with one standard deviation ($n = 70$–262). Statistically significant differences are described in the text

**Table 3 C and N assimilation associated to *Skeletonema* and *Chaetoceros***

| Diatom | POC% | PON% | C assimilation% | NO$_3$ assimilation% | NH$_4^+$ assimilation% |
|---|---|---|---|---|---|
| *Skeletonema* (still condition) | 9 | 12 | 2[a] | 8 | 6 |
| *Skeletonema* (turbulent shear) | | | 4[a] | 6 | nd |
| *Chaetoceros* (still conditions) | 27 | 38 | 10 | 38 | 35 |
| *Chaetoceros* (turbulent shear) | | | 15 | 32 | nd |

POC: Particulate organic carbon, PON: particulate organic nitrogen
POC, PON, C-, nitrate assimilation and ammonium assimilation in % of the total phytoplankton community under still conditions and turbulent shear
[a]Differences in cell-specific rates were statistically significant between still conditions and turbulent shear (two-way ANOVA, DF: 598; $p < 0.0001$)

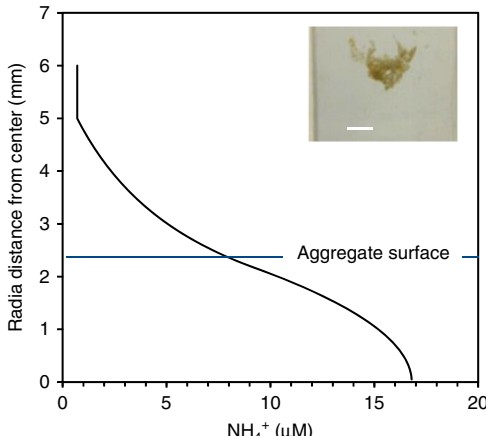

**Fig. 6** The radial distribution of ammonium concentrations in sinking diatom aggregates. The concentrations were modeled from the measured ammonium release to the ambient water. An example of such an aggregate is shown in the picture. The white size bar is 3 mm

ambient water, the ammonium production rate was similar to the ammonium assimilation rate showing that the plankton community efficiently assimilated the ammonium produced (Table 2).

## Discussion

Our knowledge and understanding of physical-biological coupling and eco-physiology at the single-cell level within mixed field populations and communities have been largely limited due to technical constraints. The recent introduction of SIMS in biological oceanography has revealed a high variability of physiological rates at the single-cell level across field populations of various (micro)-organisms[23–27]. Our study demonstrated that C and nitrate assimilation can vary by more than one order of magnitude among single cells in diatom populations. However, average values of the assimilation rates stabilised at a representative mean value of each population when >50 cells were analysed (Fig. 7). Nitrate assimilation rates co-varied with C-assimilation rates in the majority of cells (Fig. 3). A similar large variability and co-variation of C-fixation and N$_2$-fixation or NH$_4^+$ assimilation has been observed in field populations of cyanobacteria[24,27] and other phototrophic bacteria[23]. Intra-species variations in eco-physiology and nutrient demand at the cellular level provide plasticity for a population to adapt to regional heterogeneity and environmental constraints on a short time scale, as recently demonstrated at a single-cell level in cultures of heterotrophic N$_2$-fixing bacterial populations[29]. Genetic diversity within a population composed of various strains is ultimately also a prerequisite for evolution on a long time scale.

The combination of SIMS analysis with stable isotope tracers, and sensitive fluorometry enabled us to directly examine three

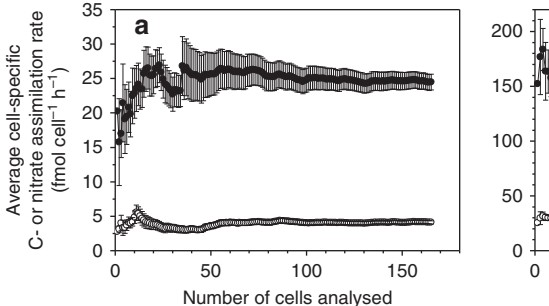
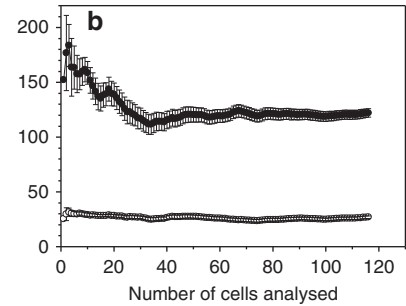

**Fig. 7** Average C- and N-assimilation rates as a function of cells analysed. Average cell-specific C-assimilation rate (closed symbols) or nitrate assimilation rate (open symbols) with the standard error of the mean as a function of cells analysed in *Skeletonema* (**a**) and in *Chaetoceros* (**b**) under turbulent shear

basic hypotheses of cell-specific C and N assimilation by chain-forming diatoms in natural mixed communities at low-nutrient concentrations in the sea. Net C assimilation in both diatom species increased with turbulent shear as expected from mass transfer theory, but surprisingly, no concurrent significant increase of nitrate assimilation was detected in the same cells as a response to turbulent shear. Contrary to predictions by mass transfer theory, large *Chaetoceros* cells assimilated N 70–94% faster than the smaller *Skeletonema*. Contributions of *Skeletonema* to C and N assimilation were modest, although its cell abundance was close to its maximum during this spring bloom[30]. Chl. *a*, suspended POC, and PON concentrations were typical of those occurring at the end of diatom spring blooms when nutrients are depleted in temperate coastal zones[5]. The larger, but less abundant *Chaetoceros* grew significantly faster than *Skeletonema* and contributed substantially to total C, nitrate, and ammonium assimilation by the entire phytoplankton community. Similar observations using nanoSIMS were recently reported in phototrophic microbial populations, where the largest, but less abundant phototrophic species contributed most to C and N assimilation in field communities[23,27]. Hence, smaller organisms do not always grow faster than larger ones do, even when nutrient concentrations are low in the field, and less abundant genera may dominate the total community production.

In our experiments, a relatively high shear rate was applied to enable significant C and N assimilation changes at the single-cell level where variability is large. The applied shear rate ($20\,s^{-1}$) corresponds to that close to the wave-breaking zone of the upper ocean[31]. Under these conditions, turbulent shear increased the net C-assimilation rate up to 59% within the longest diatom chains of *Chaetoceros* and by 31% in *Skeletonema* at low-nutrient concentrations compared to still conditions. The fluxes of nutrients with lowest molecular diffusion coefficients are generally most sensitive to turbulent shear, e.g. inorganic P fluxes are more sensitive to turbulent shear than those of inorganic N (Eq. (2))[9,19]. At a shear rate of $1\,s^{-1}$, often considered to represent the average of that occurring within the euphotic zone, the theoretical enhancements in net C-assimilation rates due to turbulence at P-limitation are 14 and 9%, while they are 9 % and 6 % at N-limitation in *Chaetoceros* and *Skeletonema*, respectively. Hence, the predicted effect of turbulent shear may also be detectable below the wave-breaking zone, and it becomes substantial when integrated over a few days. *Skeletonema* cell chains are long, thin and stiff while those of *Chaetoceros* are shorter, but thicker and more flexible. Thus, the different responses of cell-specific C-assimilation rates to turbulent shear within chains of *Chaetoceros* and *Skeletonema* may be explained by their different dimensions and morphology[32].

DIC fixation in *Skeletonema* and *Chaetoceros* is largely supplied by bicarbonate uptake (85% of total DIC fixation) during photosynthesis[33,34]. In the ocean, bicarbonate concentrations are generally in the millimolar range whereas nutrient concentrations are in the nano- to micromolar range. Diffusion-limited fluxes of DIC and nutrients to cells are proportional to their respective concentrations in the ambient water (Eq. (1)), and the modeled DIC fluxes were three orders of magnitude higher than the measured net cellular C-assimilation rates. Hence, DIC fluxes do not limit photosynthesis in many diatoms[33,34], although gradients in pH, $CO_2$ and bicarbonate concentrations occur at the cell–water interface[35]. Gross DIC fixation often exceeds C demands relative to nutrient uptake and cellular growth under nutrient limitation in chain-forming diatoms. A fraction of newly fixed carbon is excreted as dissolved organic carbon (DOC) and transparent exopolymer particles (TEP) while net C assimilation in biomass reflects nutrient-limited growth[36]. The average nitrate assimilation rate in *Chaetoceros* was similar to the upper limit determined by diffusion-limited supply through the DBL, but any short-term response in nitrate-assimilation rates to turbulent shear was masked by a high variability at the single-cell level. This may reflect storage and release of nitrate from the vacuole within the cells[37] as well as concurrent ammonium uptake. A (co)-limitation by P may also have occurred as indicated by the modeled P uptake. Nutrient limitation and uptake were thus more complex than described by diffusion limitation of one single nutrient.

Diffusion-limited nutrient uptake by individual phytoplankton, including chain-forming diatoms and other colony-forming phytoplankton, has previously been studied only using cultures and/or theoretical approaches[8–13]. In cultures, measured maximum cell-specific N-uptake rates were 53 fmol N $cell^{-1}\,h^{-1}$ and 29 fmol N $cell^{-1}\,h^{-1}$ in *Chaetoceros calcitrans* and *Skeletonema* sp. respectively, and a half-saturation constant of 0.4 µM for nitrate uptake has been reported in *Skeletonema* cultures[10]. In the present study, the average net nitrate assimilation rate within the field population of *Chaetoceros* cells was 37 fmol N $cell^{-1}\,h^{-1}$ and close to the theoretical limit constrained by diffusion limitation at a concentration of 300 nM nitrate in the ambient water (Eq. (2)). By contrast, it was 5.5 fmol N $cell^{-1}\,h^{-1}$ and ca. 5 times lower than that predicted under diffusion limitation in *Skeletonema* cells. Thus, the physiological state of *Skeletonema* may have been in the transition to stationary growth phase in the field population[30]. Carbon and nitrate, however, were assimilated close to or slightly below the Redfield ratio in both *Skeletonema* and *Chaetoceros* under still conditions, and some of the nitrate was presumably stored in the vacuole[37]. Cell-specific ammonium assimilation rates in non-aggregated *Skeletonema* and *Chaetoceros* in the ambient water were only 16% of the nitrate assimilation rates during daytime. In *Chaetoceros*, the rate increased by 39% during night compared to that measured during day (Students *t*-test: $p < 0.01$; $n = 138, 129$; Supplementary Table 1), while it increased by 86% during night in *Skeletonema* (Students *t*-test: $p < 0.01$; $n = 93, 44$; Supplementary Table 1). Thus, nitrate

assimilation dominated over ammonium assimilation in these diatoms, which did not appear to be N-limited judged by their C: N-assimilation ratios. In contrast, the ratio of C to nitrate plus ammonium assimilation in the total phytoplankton community was 8 to 10 and thus well above the Redfield ratio. Hence, N-limitation of community primary production was apparently associated with other organisms that presumably lacked nitrate reductases and were limited by available ammonium[38].

Despite its high C and N growth rates, the cell abundance of *Chaetoceros* sp. was lower than that of *Skeletonema*. Large cell size and long spines facilitate the formation of fast-sinking aggregates in *Chaetoceros* as confirmed by our laboratory experiments. In contrast, *Skeletonema* is well known to require relatively high cell concentrations for aggregate formation compared to other diatoms[5,36]. Turbulent shear stimulates aggregate formation in chain-forming diatoms, and small aggregates can form within a few hours while milimeter-sized aggregates may need 12 to 24 h to form depending on cell concentration, cell size and chain length, as was also the case in the present study[4,5,36,39]. Silicified frustules in diatoms act as biogenic ballast minerals, and sinking velocities of diatom aggregates observed in situ are on the order of 100 to 200 m d$^{-1}$ or even faster in the laboratory[2,40]. Such aggregates are readily colonised by bacteria and protozoa that partly respire and remineralize the organic matter during their export out of the euphotic zone, linking processes in the upper ocean with those in the mesopelagic ocean[6,7,41,42]. Ammonium production in aggregates of the present study was likely driven by bacterial hydrolysis and remineralization of dead diatoms and other organic matter in the aggregates[3,6,7,41]. The half-life of organic C in aggregates during microbial hydrolysis and respiration is usually one to two weeks in the upper ocean at 15 °C but increases ca. 3 fold with decreasing temperature down to the 4 °C prevailing in the mesopelagic ocean[40–42]. Sinking aggregates are therefore important vehicles of organic C and nutrient transport in the ocean.

Our study demonstrates that chain-forming diatoms link small-scale $CO_2$ and nitrate assimilation as well as ammonium production in the euphotic zone to large-scale biogeochemical fluxes of C and N in the ocean interior through the formation of aggregates. Field populations of chain-forming diatoms had a high capacity to grow and assimilate nitrate at ambient concentrations of <0.3 μM. Turbulent shear increased C assimilation and growth in the longest and largest cell chains, which were relatively low in abundance. Larger cells and cell chains, however, are more efficiently exported out of the euphotic zone compared to smaller cells and cell chains because larger chains coagulate to form aggregates at lower cell abundance than smaller ones do. Thus, turbulence is instrumental for $CO_2$ sequestration by large, chain-forming diatoms through simultaneous stimulation of cellular growth and formation of fast-sinking diatom aggregates, which drive biological $CO_2$ sequestration in the ocean[42,43].

## Methods

**Study area**. All water samples were collected at the sea surface at monitoring station B1 (N 58° 48' 18, E 17° 37' 52) of the Swedish National Monitoring Program (Swedish Meteorological and Hydrological Institute, SMHI). At the time of sampling salinity was 6 and the water temperature was 3 °C. Water analyses and experiments were performed at Askö laboratory.

**Chlorophyll *a* analysis**. Three replicates each of 500 mL of seawater were filtered onto GF/F filters. Chlorophyll *a* was extracted in 90% methanol at 4 °C during 6 h, and analysed on a spectrophotometer (Pharmacia KLB Ultrospec III)[44].

**Nutrient analysis**. Sea water was filtered through 0.2 μm syringe filters into 5 mL plastic vials and frozen until analysis. Silicic acid, nitrate, ammonium and phosphate concentrations were quantified with a continuous-flow automated system

(TRAACS 800, Bran-Luebbe, Germany)[45]. Ammonium concentrations were analysed immediately after sampling (see below).

**Microscopy**. Lugol-preserved 150 mL seawater samples for cell counting were examined under an inverted microscope (Zeiss, Axio) at ×200 magnification after the cells had settled in a 25 mL Utermöhl sedimentation chamber (Hydrobios, Germany). Number of cells, cell size and chain length of the dominant species (*Skeletonema* and *Chaetoceros*) were quantified. Cell volumes were calculated as cylinders: $V = \pi r^2 h$.

**Mass transfer theory**. The potential diffusive DIC and nutrient supply to cells within cell chains were calculated from the analytical solutions of diffusion to a cylinder under still conditions[46]:

$$Q_t = \left[8 + 6.95\left(\frac{L}{D}\right)^{0.76}\right] r_0 D(C_\infty - C_0) \tag{1}$$

where $Q_t$ is the quantity of substance diffusing to a cell within the chain with the length ($L$) and diameter ($D$) per unit time, $t$, $D$ is the diffusion coefficient of the substance, $C_0$ its concentration at the cell surface, $r_0$ and $C_\infty$ the concentration in the ambient water. The equation is considered accurate for $L/D < 8$. We assumed $C_0$ to be zero, and used the molecular diffusion coefficients for $HCO_3^-$ ($6.0 \times 10^{-6}$ cm$^2$ s$^{-1}$), $NO_3^-$ ($9.7 \times 10^{-6}$ cm$^2$ s$^{-1}$) and ortho-phosphate ($3.6 \times 10^{-6}$ cm$^2$ s$^{-1}$) at 3 °C and a salinity of 6 as previously reported[47].

The Sherwood number, Sh, describes the relative increase in transport-limited fluxes due to fluid motion in the vicinity of a cell as compared to that at still conditions[22]. It depends on the Peclet number, Pe, and can be calculated as a function of shear rate due to turbulence for $0.1 < Pe < 100$[9]:

$$Sh = 1.002 + 0.21\left(\frac{Er_0^2}{D}\right)^{1/2} \tag{2}$$

where $E$ is the shear rate (s$^{-1}$) and ($Er_0^2/D$) is Pe.

In our experiments, we used couettes to create laminar shear[36] with a shear rate of 21 s$^{-1}$ in order to measure statistically significant differences in nutrient uptake under still and under turbulent conditions (please see below). Pe numbers were 1 and 3 for *Skeletonema* and *Chaetoceros*, respectively, when calculations were based on the equivalent spherical radius of the colony volume. The average equivalent spherical radius of cell chains was 7.9 μm and 12.2 μm for *Skeletonema* and *Chaetoceros*, respectively.

**Incubations with stable isotope tracers**. The phytoplankton community was incubated with stable isotope tracers ($H^{13}CO_3^-$ combined with $^{15}NO_3^-$ or $^{15}NH_4^+$). The final excess $^{13}C:^{12}C$ isotope ratio of $H^{13}CO_3^-$ in the ambient water was 3.6% combined with 5.6% excess $^{15}N:^{14}N$ isotope ratio of $NO_3^-$ or 80% excess $^{15}N:^{14}N$ isotope ratio of $NH_4^+$, respectively. Final concentrations of $NO_3^-$ and $NH_4^+$ were similar (0.3 μM). Bottles were carefully shaken after addition of the stable isotope tracer. For the quantification of $NH_4^+$ assimilation and $NH_4^+$ production, nine 1 L bottles were incubated by dual labelling with $H^{13}CO_3^-$ and $^{15}NH_4^+$ and one control without tracers in natural light in an outdoor mesocosm at in situ temperature at noon. The control bottle was used to measure the natural isotope ratios of $^{13}C:^{12}C$ and $^{15}N:^{14}N$ in the diatoms cells as well as in the phytoplankton community. An additional set of bottles were incubated around midnight. Triplicate incubation bottles were stopped by filtration at $T_0$ and after 2 and 5 h. For the quantification of nitrate assimilation, the phytoplankton community was incubated with $H^{13}CO_3^-$ and $^{15}NO_3^-$ in 1 L Duran bottles under still conditions or under laminar shear in couettes[36] in the laboratory. Three Duran bottles and three couettes with stable isotope tracers were incubated for 8.5 h in a thermostated room at in situ temperature and a light intensity of 220 μmol m$^{-2}$s$^{-1}$. Three 1 L Duran bottles and one couette without stable isotope tracers served as controls. All Duran bottles were placed horizontally next to the couettes, which were operated with a speed of 0.9 revolutions s$^{-1}$.

Samples were fixed at 4°C for 24 h with 2% paraformaldehyde (PFA) for SIMS analysis of C and N assimilation by *Skeletonema* sp. and *Chaetoceros* sp. and washed after filtration onto GTTP filters (pore size 0.22 μm; diameter 25 mm, Millipore, Eschborn, Germany). Furthermore, 500 mL samples were filtered onto pre-combusted GF/F filters for elemental analysis-isotope ratio mass spectrometry (EA-IRMS) analysis of C and N assimilation by the whole phytoplankton community (see below). Twelve millilitre samples were filled, headspace free into gas tight Exetainer©vials to which 100 μL 50% ZnCl$_2$ solution was added for later analysis of $^{13}C:^{12}C$-$HCO_3^-$, $^{15}N:^{14}N$-$NO_3^-$ and $^{15}N:^{14}N$-$NH_4^+$. $^{13}C$-$^{12}C$ isotope ratios of bicarbonate were analysed at Stable Isotope Facilities, University of California Davis, USA, and the $^{15}N:^{14}N$-$NO_3^-$, and $^{15}N:^{14}N$-$NH_4^+$ isotope ratios were analysed at the Stable Isotope Laboratory Stockholm University (see below).

**Elemental analysis-isotope ratio mass spectrometry**. GF/F filters were freeze-dried and decalcified overnight in fuming 37% HCl in a desiccator prior to analysis by a Thermo Flash EA 1112 elemental analyzer coupled to an isotope ratio mass spectrometer (Thermo Delta Plus XP, Thermo Fisher Scientific). Caffeine was used

as standard for isotope correction and C/N quantification of bulk C and N assimilation and bulk C and N assimilations rates were calculated as previously described[48]. EA-IRMS analysis was completed before SIMS analysis was initiated.

**Secondary ion mass spectrometry analysis**. We used a large-geometry SIMS instrument (Cameca IMS 1280; Gennevilliers, France) at the NORDSIM facility, Swedish Museum of Natural History. The IMS1280 is a dynamic SIMS, with magnetic sector mass separation, and imaging capability similar to nanoSIMS. However, the spatial resolution of IMS1280 is ca. 1 μm whereas that of nanoSIMS is ca. 50 nm. The lower spatial resolution of IMS1280 allows for higher through-put compared to nanoSIMS when cells are large (>3 μm). Prior to analysis, the filters containing chemically fixed cells were coated with a nanometer thin gold layer and cut into ca. 4×4 mm$^2$ pieces and mounted on a sample holder. Analysis was performed using a Cs$^+$ primary beam with a spatial resolution of ca. 1 μm. We only analysed dispersed cell chains on the filters. Diatom cell chains were pre-sputtered with a beam of 10 nA for 5 min to remove the silicified cell wall, and then imaged using a 40–80 pA primary beam for 100 cycles. The pre-sputtered area was larger than the imaged area in order to eliminate possible slight offsets between the two beams (sputter and analytical) and edge effects. Analyses were automated into cells after a fixed period of pre-sputtering needed to reach the interior of cells with high CN had been determined. For each cell chain we recorded secondary ion (SIMS) images of $^{13}C^{14}N^-$ and $^{12}C^{14}N^-$, and $^{12}C^{15}N^-$ using a peak-switching routine at a mass resolution of ca. 6000 ($M/\Delta M$), which is less than the ~12,000 $M/\Delta M$ necessary to resolve BO$^-$ from $^{13}C^{14}N^-$[47]. BO$^-$ is a common surface contaminant and present as a trace element in the diatom frustules[49,50]. Also, ~7000 $M/\Delta M$ is necessary to fully resolve $^{12}C^{14}N^-$ from $^{13}C^{13}C^-$, which can be significant for highly $^{13}C$-enriched samples. However, measurements on control samples suggest that these interferences did not significantly affect the isotope measurements inside the cells, but BO$^-$ slightly affected ratios in low yield regions, e.g. the frustules. The use of CN isotopomers have the advantage that they are bright and two isotope ratios can be measured from measurements of three isotopomers. The imaged area was 80×80 μm with a scanning ion imaging of 256×256 pixels. The dwell times were 1 s, 5 s and 2 s for $^{12}C^{14}N$, $^{12}C^{15}N$ and $^{13}C^{14}N$, respectively (wt time 0.8 s, 100 cycles). Isotope ratio data were extracted from individual cells based on cell morphology visualised in the $^{12}C^{14}N$ ion images. Image and data processing were performed as previously described for nanoSIMS analysis using the CAMECA software[23–26]. Each image of each cell was carefully examined by eye and the region of interest (ROI) was defined along the border of the $^{12}C^{14}N$ image and drawn by hand using the CAMECA software. The $^{13}C/^{12}C$ and $^{15}N/^{14}N$ ratios of 572 individual *Skeletonema* cells and of 306 individual *Chaetoceros* cells were determined for samples incubated with $^{13}C$ and $^{15}N$ tracers, whereas $^{13}C/^{12}C$ and $^{15}N/^{14}N$ ratios of 102 cells were determined from incubations without isotope tracers (controls). The small interference by $^{11}B^{16}O$ isotopes from the silicate frustule at a mass resolution of 6000 ($M/\Delta M$) was corrected for when subtracting the natural apparent signal from $^{13}C/^{12}C$ in un-amended samples (controls) from that of amended samples. The average $^{13}C/^{12}C$ in un-amended samples (controls) was 0.0113 ± 0.0006 (stdev; $n = 102$), whereas that of amended samples was 0.0143 ± 0.0017 (stdev; $n = 878$).

**C and N growth rates by cells**. C-specific C-assimilation rates and N-specific N-assimilation rates, $k$ (h$^{-1}$), in the cells were calculated from the excess $^{13}C/^{12}C$ and $^{15}N/^{14}N$ isotopic ratios (IR) in individual cells relative to those measured in the un-amended controls, the excess %—isotopic ratios of bicarbonate or inorganics nitrogen in ambient water ($F_{bulk}$), and the incubation time ($t_1-t_0$) with stable isotopic tracers[24–26]:

$$k\left(\mathrm{h}^{-1}\right) = \frac{\mathrm{IR}_{t_1} - \mathrm{IR}_{t_0}}{F_{bulk}(t_1 - t_0)} \qquad (3)$$

C and N content of diatom cells were calculated from their dimensions measured by light microscopy in parallel samples[28]. Doubling time was calculated as ln2/$k$. Cell-specific rates were calculated by multiplication of $k$ with the cellular C or N content.

**Ammonium analysis**. Total NH$_4^+$ concentration in the ambient water was measured in triplicate on a Trilogy® Laboratory Fluorometer (Turner Designs, USA)[51]. Triplicate subsamples for isotope analysis were transferred into 12 mL Exetainer©vials and biological activity stopped by adding 100 μL 50% ZnCl$_2$ solution. The ratio of $^{15}N/^{14}N$-NH$_4^+$ was analysed after helium de-gassing to remove $^{15}N_2$ and $^{14}N_2$ and subsequent chemical conversion of NH$_4^+$ to N$_2$ with alkaline hypobromite (NaOBr)[52,53]. Isotopic ratios of $^{28}N_2$, $^{29}N_2$ and $^{30}N_2$ were analysed by gas chromatography isotope ratio mass spectrometry on a Thermo Delta V isotope ratio mass spectrometer. NH$_4^+$ utilisation, assimilation and production rates were calculated from changes in $^{15}NH_4^+$ and $^{14}NH_4^+$ concentrations over time and the change in the isotope composition of particulate matter analysed by EA-IRMS[54]. The $^{15}NH_4^+$ concentration decreased exponentially over time, and $^{15}NH_4^+$ labelling percentage was averaged over the incubation period.

**Ammonium production in aggregates**. Seawater samples were incubated in six 1.5 L roller tanks with a rotation speed of 3 rpm within an hour after sampling[39]. After 24 h, 24 aggregates were collected from the roller tanks and incubated individually in Falcon tubes with 0.2 μm filtered sea water (and six controls without aggregates)[6]. Ammonium concentrations in ambient water of the vials containing aggregates were significantly higher than those in the controls (Student's $t$-test: $p < 0.05$; $n = 6$, 24). The ammonium distribution within and around aggregates were modeled as previously described[6].

**Statistics**. The statistical significance of differences in measured carbon and nitrogen assimilation rates at the community level (EA-IRMS data) were analysed using two-way ANOVA test in SigmaPlot version 11. The replicates of cell-specific C and N-assimilation rates measured by SIMS were selected from the bottles with community nitrate assimilation rates closest to the mean value the nitrate assimilation measured by EAIRMS under still conditions and turbulent shear, respectively. Increasing numbers of the cellular C and N-assimilation rates measured by SIMS were calculated until the average values per cell were stable and the standard error was <5% of the average value to achieve representative average values for single cells of both genera of diatoms (Fig. 7). The C and N-assimilation rates calculated from the $^{13}C/^{12}C$ and $^{15}N/^{14}N$ ratios of the 572 individual *Skeletonema* cells and of the 306 individual *Chaetoceros* cells, and the 102 control cells were used in the statistical analysis.The C-assimilation rates followed a normal distribution as tested in detail using statistical analysis software (SAS) of their residuals. The C and N-assimilation data in the sub-groups of cells were also described by normal distributions and (non)-significant differences between these were analysed by two-way ANOVA test in SigmaPlot version 11 as well as by Student's $t$-test in Microsoft Excell (2010). These two different statistical analyses gave similar results.

**Data availability**. The data that support the findings of this study are available and can be acquired from the corresponding author on request.

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

## Acknowledgements

We thank the staff at Stockholm University's Baltic Sea Centre for their hospitality and support during this study at Askö Laboratory. The NordSIM ion microprobe facility is operated under an agreement between the research funding agencies of Denmark, Iceland, Norway and Sweden, the Geological Survey of Finland, and the Swedish Museum of Natural History. This is NordSIM publication # 563. This study was supported by University of Gothenburg, and the Swedish Research Council (VR, Dnr: 621-2011-4406 and Dnr: 2015-05322 to H.P.), and by the Baltic Ecosystem Adaptive Management Programme www.su.se/beam) to H.P., V.B. and I.K. We are grateful to Helena Höglander for contributing with the image of the plankton community shown in Fig. 1. Gabriele Klockgether kindly analysed EA-IRMS samples at the Max Planck Institute for Marine Microbiology in Bremen, Germany. We thank Kerstin Wiklander for statistical analysis and advice. We also thank Lars Arneborg for discussions, and Erik Selander, Hans-Peter Grossart and Eva-Maria Zetsche for constructive comments on an earlier draft of this manuscript, and Pete Jumars for his insightful comments and thorough review of the manuscript.

## Author contributions

J.B., I.K. and H.P. designed and carried out the experiment. J.B., I.K., M.J.W., G.L., V.B. and H.P. contributed to sample and data analysis, and writing of the manuscript.

## Additional information

**Competing interests:** The authors declare no competing interests.

