## [Peer Review File · Nature Communications]

Editorial Note: In their review of the first version of this manuscript, reviewer 1 added their comments to the manuscript file. These comments, excluding minor textual revisions, have been copied into this Peer Review File.

Reviewers' comments:

Reviewer #1 (Remarks to the Author):

I am very excited to see experimental evidence of turbulence effects on nutrient uptake at the single-chain and single-cell level for the first time for field-collected phytoplankton. I am not an expert on the isotopic methods and so cannot give critical review on those aspects, but they appear to be very innovative and to hold much promise for future work. There is a serious problem in the theoretical context provided by the authors, however, that needs to be corrected before a second review.

Lines 438-440—I got as far as Equation 2, followed by the sentence “We assumed r_1 / r_0 to be 10, i.e., that the concentration boundary layer thickness was 10 times larger than the cell radius.” This choice is arbitrary and is clearly a poor one for this problem. Flux to a sphere in still water is an analytically soluble problem, and at $r = 10$ times cell radius concentration has only reached 90% of that at infinity. Flux to an infinite cylinder is well known to be an analytically insoluble problem unless concentration at some distance from the cylinder is given. Concentration gradients in a cylindrical diffusion system must be shallower, not steeper, than gradients in an otherwise comparable, spherical system, however, so the choice made is certainly wrong and will overestimate. There is no way I know of to make any particular choice for a still-water solution, but 100 or 1000 is arguably better. The authors don't specify the concentration at the cell surface, so I assume they used zero (perfect absorption). Explicit specification would be best. Eq. 2 has two shortcomings in this application. The first is the inability to specify r_1 objectively. The second is that chemical gradients are steeper near the ends of a finite cylinder. Choosing an unreasonably small value for r_1 compensates partially for the second problem.

Fortunately, empirical solutions are available for some finite cylinders. On p. 89, Clift et al. (1978) give a solution

as: $\left[8 + 6.95 \left(\frac{\text{length}}{\text{diameter}} \right)^{0.76} \right] \text{radius}$. The range, however, extends only up to length/diameter = 8 so it will work (accurately) for only one of the chain species. The calculated quantity needs to be multiplied by the diffusion coefficient and the difference between surface concentration and far-field concentration. I've extended the solution numerically to a length/diameter ratio of 100, and express it instead as the ratio of flux for a cylinder relative to that for a sphere of the same radius:

I was frustrated in trying to reproduce calculations by two factors. One is the seemingly random choice of units (not uniformly SI). Scientific notation and uniform mks usage would be far preferable. The other is lack of sufficient description (or ambiguity in the description) to make some of those calculations. The text says that chains were approximated as cylinders, but then fluxes are given per cell. Some of the ambiguity arises from using the term “cell chains” (lines 392 and 438). To check whether I could reproduce the stated fluxes, I calculated the flux using Eq. 3 for nitrate and *Skeletonema* using the r_1 value that the authors did. Converting all the variables to SI:

$$\frac{\text{cylinder flux}}{\text{sphere flux}} = \frac{2}{\pi} + 0.0748E + 0.493E^{0.694}, \text{ where } E \text{ is the length/diameter of the cylinder.}$$

$$\frac{2\pi D(C_1 - C_0)}{\ln \frac{r_1}{r_0}} = \frac{2\pi(105 \times 10^{-6})(9.78 \times 10^{-10})}{\ln \frac{25 \times 10^{-6}}{2.5 \times 10^{-6}}} = 2.80 \times 10^{-13}$$

. I used C_1 and not C_∞ because it is the concentration at the

non-infinite distance r_1 . The solution has units of moles m^{-3} . Converting to the mixed units used on p. 8 yields $0.282 \text{ pmol h}^{-1}$. Dividing by the mean number of cells per chain gives $0.040 \text{ pmol cell}^{-1} \text{ h}^{-1}$. That's close, but 11% lower than the figure given on p. 8. I don't know where the difference arises. The diffusion coefficient I used was for 0°C from reference 46. If I make a linear interpolation to 6°C , I overshoot ($0.049 \text{ pmol cell}^{-1} \text{ h}^{-1}$) the authors' result. The point is that the description is not quite explicit enough to duplicate the calculations. For example, I assumed that the authors assumed $C_0 = 0$, but they never stated that assumption.

Using the cylinder-flux/sphere-flux ratio gives a solution of $0.028 \text{ pmol cell}^{-1} \text{ h}^{-1}$ 63% of the authors' figure for *Skeletonema*. For *Thalassiosira*, the result is $0.041 \text{ pmol cell}^{-1} \text{ h}^{-1}$. This result is closer (74%) because of the smaller aspect ratio and hence steeper gradients than around an infinite cylinder with $r_1/r_0 = 10$.

Lines 39-40—Clarify whether the 24% is of the total or in addition to the total for ambient water. (I learned on p. 9 that it was the latter.)

Line 101—Although it is true that viscous dissipation produces heat, it is the shear and not the heat that erodes chemical boundary layers and enhances fluxes. Please clarify.

Line 102—The cited reference (9) does not say that micrometer-sized phytoplankton benefit from shear-produced thinning of their diffusive boundary layers because it is not true. The boundary layer on a cell this small extends only a few micrometers, a scale over which diffusion vastly exceeds advection in delivery speeds. The reference cited hypothesizes that cells $60 \mu\text{m}$ or more in radius can get substantial increases in arrival fluxes of nutrients from the levels of viscous shear produced by decaying turbulence. Somewhere early on, the authors should state whether they are using equivalent spherical radii or equivalent spherical diameters.

Line 262—Some may not consider 5 and 8% to be “considerable.” A percentage that small may be exceedingly difficult to detect in the face of natural variability.

Lines 440 - 441—Elsewhere (line 454) you say $7.0 \times 10^{-6} \text{ cm}^2 \text{ s}^{-1}$ was used for all the ions. Clarify whether you used different coefficients elsewhere but used only one for all Sherwood number calculations.

With a better theoretical context (for diffusion to finite cylinders), I think this manuscript could be a strong contribution.

While the order is not incorrect grammatically, "spring diatom bloom" is a much more familiar wording

The original phrasing suggests that heat enhances diffusion. Although viscous shear produces heat, the temperature change is not sufficient in the ocean to significantly enhance diffusion.

There is no way that realistic levels of shear can significantly increase flux to a $1 \mu\text{m}$ diameter cell. Diffusion times over the tiny diffusional boundary layer are simply way too short. Reference 9 does not make this claim, instead suggesting that significant enhancement will occur only for cells larger than $60 \mu\text{m}$ radius. One confusing issue is that some authors use radius whereas others use diameter. I suggest stating the scale used and sticking with it throughout.

Sincerely yours,

Peter A. Jumars
Professor Emeritus of Oceanography
jumars@maine.edu

Reviewer #2 (Remarks to the Author):

This study seeks to provide single cell data to test the hypothesis that that primary production in chain-forming diatoms is stimulated by turbulence. I was asked to review the application of SIMS in this study, and therefore most of my comments are from that perspective.

I found this a poorly organized and incompletely presented manuscript, and therefore it was very difficult to evaluate, even within my limited scope. The SIMS results are not well organized and are hard to reconcile with each other, particularly on the point of C-assimilation by *Chaetoceros*. The figures lack basic reference information, and Fig. 4 in particular is hard to interpret and reconcile with the narrative. The Method Section does not include important information and presents the work as centered on two experiments, which is not how the research is presented in the Results. Finally, the conclusion that turbulence stimulates small-scale biological CO₂ assimilation (L40-41) does not seem to follow from the results: the narrative associated with Fig. 4 states that there is not a turbulence treatment effect for C-assimilation by *Chaetoceros* (L149-151), though the cell chain results make that result hard to understand (L185-187). I did not analyze the N-assimilation results in detail given the substantive problems with the manuscript.

Methodologically, I was concerned that the statistical methods were not discussed in the Method section, particularly because it was not clear how the replicates were handled. I was also concerned about standardization of the measurements, use of the data for the control samples, and measurement precision. Finally, based on the SIMS Methods section, the mass resolving power used was not sufficient to resolve what is typically a significant isobaric interference at 27 amu (11B 16O- on 13C 14N-), which would affect the 13C/12C measurement. This interference is likely the cause of the background in the 13C/12C images (see below), and therefore the TEP interpretation should be reconsidered. BO- would also be an interference in the measurements on the cells, but it is potentially not significant when the CN- count rate is high. Standard and control measurements are necessary to demonstrate that the cellular 13C/12C data are valid. These issues will need to be considered in the statistical analysis. If there were other 13C/12C measurements (e.g., using C dimers or monomers), that could be useful.

As a general point, based on the method section, there were bulk IRMS data for these experiments, but I could not find them in the manuscript. Given the variability in the SIMS data, bulk C and N assimilation data would be a useful reference for the single cell data.

My detailed comments follow:

L66: The description of SIMS does not allow for differences among SIMS instruments. It would be better to be more specific about the critical properties for this study (e.g., imaging, dynamic, magnetic sector).

L134: The use of the term “stable isotope” in context of the images implies that these images are different from other secondary ion images, which they are not. I suggest referring to the image data as ion images and ion ratio images or possibly SIMS images.

L135: I suggest giving a basic statement on how the image data are used: e.g., isotope ratio data are extracted from individual cells based on cell morphology visualized in the CN ion images.

L135-138: The explanation of the difference in relative ^{15}N and ^{13}C enrichment in the TEP is not satisfactory. These ratios reflect the source of the N and C (here, old versus new), not the concentration of each element in the material. Taken on its face, higher ^{13}C enrichment relative to ^{15}N enrichment suggests that the C is new and the N is not. However, based on the reported mass resolving power for the analyses and the species monitored, it is likely that the area around cells in $^{13}\text{C}/^{12}\text{C}$ images reflects a background count rate of $^{11}\text{B}^{16}\text{O}^-$ at 27 amu (see below). This is even more likely since the CN $^-$ count rate in the region around the cells is very low (i.e., BO $^-$ counts are more likely to be significant). It would be useful to have data for a reference material, such as the substrate around unlabeled diatoms, to rule out this possibility.

If this is an interference issue, it would not necessarily make the data set for the cells (high CN $^-$ regions) useless, but the uncertainty on the measurements would have to be re-evaluated.

If the result stands, I would suggest that this discussion not come as the first note on the images since this is an interpretation. I also want to note that I do not think that it is obvious that the images can be directly interpreted for relative enrichment. To the extent that the authors discuss their interpretation of the TEP enrichment, I think it would be better to make the interpretation based on numerical data.

At the risk of causing confusion, I'll just note here that the way the authors used the word “background” in this context is confusing. The word “background” implies counts that are not from the sample, whereas the authors seem to mean counts that are not directly from the diatoms themselves, but rather from the TEP. This, however, is beside the point if BO $^-$ is the issue.

L132: Please be more explicit about the presentation of the nitrate and ammonium experiment data. Based on the Methods section, there are two sets of experiments, but this section does not organize the results in that way, making it more difficult than necessary to relate them to each other.

L143: Are the numbers mean +/- 2 standard errors of the mean?

L147: How do the replicates compare? How are they handled in the statistical test?

L167: Is "C-specific C-assimilation" common term?

L180-191: It is surprising that there is a significant treatment effect for the Chaetoceros cells in the interior of the chains, but not effect overall. Does this mean that most of the cells discussed in the previous section were primarily not in chains? At the start of that section, on L134, it is stated that the data were "measured in single cells within individual cell chains." I am guessing this is just a clarity of writing issue.

L241: I don't understand. This statement seems to contradict L149-151, which state that there was no statistical difference in cell-specific C-assimilation rates for Chaetoceros .

L280-282: Unless the authors can produce a high resolution mass spectrum demonstrating that 11B 16O- was resolved from 13C 14N- at mass 27 amu, this line cannot be supported. Also, it would be more convincing if there were significant counts of the species used for the presumed TEP measurements.

L357: The meaning of "final labeling" is not clear. Does this mean 5% of the bicarbonate in the incubation was labeled? If that is the case, it is relevant to know if this is the calculated abundance of 13C, given that 13C is naturally a 1% isotope.

L361-362: Please be more precise. Presumably the authors mean: "and one 1L bottle was incubate with natural, unlabeled NH4+ at the same concentration as the other bottles..."

L363-364: If there were nine bottles and two sampling points at which times 3 bottles were sampled, what happened to the last three bottles? Was there a 3rd sampling point? (see question below: was it 2 and 5 hours?) When was the control sampled?

L364: 25 h or 2.5 h or something else?

L373: The phrasing here raises the question of whether the T0 samples had been exposed to labels, but immediately sampled, or were not exposed to label. Based on the description above, I understood that it was the latter. If that is the case, "At the start" would be a better way of stating.

L385: In the SIMS method section, no mention is made of standardization of the analyses. Please include.

L391: There is insufficient information to assess the analysis method. What is the basis for stating that silica is removed but the cellular material remains? Do they have an estimate of the depth of sputtering? What was the area sputtered with the 10 nA beam? Why "< 5min"? Why not a more specific sputtering period? Was there a specific marker used to determine that the correct depth was reached? Where the cells imaged by SEM after sputtering?

L392: What was the size of the area analyzed? What was the mode of analysis (presumably scanning ion imaging)? If scanning, what was the pixel number and dwell time?

L394: 6000 MRP is not sufficient to resolve $^{13}\text{C}^{14}\text{N}^-$ from $^{11}\text{B}^{16}\text{O}^-$, which can interfere at mass 27 amu at relatively high abundance in many samples. The mass difference between $^{13}\text{C}^{14}\text{N}^-$ and $^{11}\text{B}^{16}\text{O}^-$ is ~ 2.2 milli-atomic mass units, which requires $\sim 12,000$ MRP to resolve, which the CAMECA ims-1280 can do. Boron is a common surface contaminant and it is present in seawater. The "background" mentioned for Fig. 3 E & F may in fact be BO^- counts that have become significant relative to the $^{13}\text{C}^{14}\text{N}^-$ counts. Where the CN^- count rates are high, BO^- may not be significant. Potentially the authors can incorporate this potential source of variability into their statistical evaluation. I am concerned, however, that the number of control samples may limit their statistical power.

It would be instructive, but not necessary, to state why the CN isotopomers were used for measuring the $^{13}\text{C}/^{12}\text{C}$ ratio.

Did the team also collect C2- for $^{13}\text{C}/^{12}\text{C}$ measurements? That requires lower MRP and could potentially have been collected simultaneously.

L400-402: For the C and N assimilation rates, please cite one paper that has the exact procedure used or describe the procedure. Differences among the procedures cannot be reconciled by the reader.

L403: Based on L166, cell size is determined by SIMS images. Was this done using the data processing software? What were the criteria? 50% height of CN-? Do the authors need to correct for erosion of the lateral dimensions of the cells during high current sputtering?

L427: The source of the derivation of this equation should be cited, and the limitations should be provided. Four papers are cited above in the text, which is not sufficiently specific. Note that this equation is inaccurate at high levels of enrichment relative to the pool because of its approximations. The authors should check that their data are inside the accurate range. Popa et al. 2007, ISME J, 1: 354 has a complete derivation of a related equation without approximations.

Fig. 2: This image needs a scale bar.

Fig.3. Color scales are hard to read. What are the counts in A and B? They are relatively low for CN if that is a cumulative by pixel. How are the ROIs defined?

Fig. 4. I'm having a hard time with this figure. I am not seeing the relative changes indicated in the text. Perhaps these figures should be in the Supplemental, and instead the average and standard deviation of the data should be plotted on a single graph, along with the number of data points represented by each data point.

I find this graph confusing relative to the two experiments called out in the Methods.

Fig.5. It would be helpful to have statistical differences indicated in the graph. Also, it would be helpful to have the number of analyses represented in each bar shown in parentheses to aid reader interpretation.

Fig.6. "deviation" is misspelled. As above, indicating statistical difference and number of analyses for each bar would be helpful.

Reviewer #3 (Remarks to the Author):

Review of: Small-scale and large-scale CO₂ sequestration by chain-forming Diatoms are simultaneously stimulated by turbulence in the sea.

By: Johanna Bergkvist, Isabell Klawonn, Martin J. Whitehouse, Gaute Lavik, Volker Brüchert, and Helle Ploug

This paper describes some novel findings that have potentially consequences for our understanding of the oceans' biogeochemistry. It demonstrates, quite convincingly, that the nutrient uptake and growth of large chain-forming diatoms is enhanced under turbulent conditions; conditions under which they also form fast sinking aggregates, thus removing both carbon and nutrients from the surface ocean. The paper is well written, and should be of broad interest to marine and aquatic scientists. I recommend the publication of this manuscript, but draw the authors' attention to a few details that they should consider.

Line 93-95: While I appreciate the intent here, the points made are a bit sloppy. Firstly, I'm not sure what "shorter diffusion distance" refers to. The diffusive flux to a small object is smaller than that to a large object, scaling linearly with size. The second point that is questionable is the reference to "a

larger surface area : volume ratio” for small cells. This is true but not germane to the process. Rather it is the linear dimension: cell volume ratio that is pertinent, essentially the ratio of the diffusive flux of nutrients to the demand for growth.

Line 101: perhaps reword this – it is not the fact that turbulent energy is dissipated as heat that enhances diffusion limited exchange.

Under turbulent conditions as simulated in the in the roller tanks, there are several processes going on at once. There is the increased nutrient flux to individual cells due to shear, there is the coagulation of cells and detrital matter into aggregates with a corresponding increase in sinking speed, and there is the bacterial colonization of these aggregates with subsequent remineralization of detrital nutrients. Each one of these processes has an impact on the nutrient dynamics and growth of the diatoms. It would be really helpful for the reader if the authors could try to disentangle these processes. The central premise, surmised from Figure 1, is that increased uptake due to turbulent shear is the candidate mechanism, but can the other processes be discounted. At the very least, this should be mentioned in the discussion.

Responses to reviewers

We would like to thank all three reviewers for their constructive comments and suggestions which certainly improved the manuscript. Considering the theoretical context, we have included diffusion to a finite cylinder as suggested by reviewer#1 and clarified assumptions for all calculations. We have carefully re-examined all data concerning the BO issue as raised by reviewer #2. We found minor inconsistencies in calculations in this very large data set. These have been corrected in the revised manuscript. We have also included a new figure (Fig. 8) which better describes how we dealt with the cell-to-cell variation in our analysis. As suggested by reviewer #3, we have included more discussion on the time constraints for aggregate formation, sinking velocities as well aggregate degradation in the revised manuscript. These changes, however, did not change the major conclusions of the original submitted manuscript.

Our detailed responses can be found in the following pages:

Reviewer #1

General comments:

I am very excited to see experimental evidence of turbulence effects on nutrient uptake at the single-chain and single-cell level for the first time for field-collected phytoplankton. I am not an expert on the isotopic methods and so cannot give critical review on those aspects, but they appear to be very innovative and to hold much promise for future work. There is a serious problem in the theoretical context provided by the authors, however, that needs to be corrected before a second review.

Authors: We thank the reviewer for his enthusiastic comments and for editing the manuscript file as well! We have improved the theoretical context for diffusion to finite cylinders as suggested by the reviewer. Please see below for details.

Lines 438-440—I got as far as Equation 2, followed by the sentence “We assumed r_1/r_0 to be 10, i.e., that the concentration boundary layer thickness was 10 times larger than the cell radius.” This choice is arbitrary and is clearly a poor one for this problem. Flux to a sphere in still water is an analytically soluble problem, and at $r = 10$ times cell radius concentration has only reached 90% of that at infinity. Flux to an infinite cylinder is well known to be an analytically insoluble problem unless concentration at some distance from the cylinder is given. Concentration gradients in a cylindrical diffusion system must be shallower, not steeper, than gradients in an otherwise comparable, spherical system, however, so the choice made is certainly wrong and will overestimate. There is no way I know of to make any particular choice for a still-water solution, but 100 or 1000 is arguably better. The authors don’t specify the concentration at the cell surface, so I assume they used zero (perfect absorption). Explicit specification would be best. Eq. 2 has two shortcomings in this application. The first is the inability to specify r_1 objectively. The second is that chemical gradients are steeper near the ends of a finite cylinder. Choosing an unreasonably small value for r_1 compensates partially for the second problem. Fortunately, empirical solutions are available for some finite cylinders. On p. 89, Clift et al. (1978) give a solution as:

$$Q_t = \left[8 + 6.95 \left(\frac{L}{D} \right)^{0.76} \right] r_0 \times D \times (C_\infty - C_0) \quad (Eq.1)$$

The range, however, extends only up to length/diameter = 8 so it will work (accurately) for only one of the chain species. The calculated quantity needs to be multiplied by the diffusion coefficient and the difference between surface concentration and far-field concentration. I’ve extended the solution numerically to a length/diameter ratio of 100, and express it instead as the ratio of flux for a cylinder relative to that for a sphere of the same radius:

$$\text{cylinder flux sphere flux} = 2\pi + 0.0748E + 0.493E^{0.694}, \quad (Eq. 2)$$

where E is the length/diameter of the cylinder.

I was frustrated in trying to reproduce calculations by two factors. One is the seemingly random choice of units (not uniformly SI). Scientific notation and uniform mks usage would be far preferable. The other is lack of sufficient description (or ambiguity in the description) to make some of those calculations. The text says that chains were approximated as cylinders, but then fluxes are given per cell. Some of the ambiguity arises from using the term “cell chains” (lines 392 and 438). To check whether I could reproduce the stated fluxes, I calculated the flux using Eq. 3 for nitrate and *Skeletonema* using the r_1 value that the authors did. Converting all the variables to SI: $2\pi l D (C_1 - C_0) \ln r_1 / r_0 = 2\pi 105 \times 10^{-6} (9.78 \times 10^{-10}) \ln 25 \times 10^{-6} / 2.5 \times 10^{-6} = 2.80 \times 10^{-13}$. I used C_1 and not C_∞ because it is the concentration at the non-infinite distance r_1 . The solution has units of moles m^{-3} . Converting to the mixed units used on p. 8 yields 0.282 pmol h^{-1} . Dividing by the mean number of cells per chain gives 0.040 pmol $cell^{-1} h^{-1}$. That's close, but 11% lower than the figure given on p. 8. I don't know where the difference arises. The diffusion coefficient I used was for 0 °C from reference 46. If I make a linear interpolation to 6 °C, I overshoot (0.049 pmol $cell^{-1} h^{-1}$) the authors' result. The point is that the description is not quite explicit enough to duplicate the calculations. For example, I assumed that the authors assumed $C_0 = 0$, but they never stated that assumption.

Using the cylinder-flux/sphere-flux ratio gives a solution of 0.028 pmol $cell^{-1} h^{-1}$ 63% of the authors' figure for *Skeletonema*. For *Thalassiosira*, the result is 0.041 pmol $cell^{-1} h^{-1}$. This result is closer (74%) because of the smaller aspect ratio and hence steeper gradients than around an infinite cylinder with $r_1/r_0 = 10$.

*Response from authors: The reviewer is right that we assumed C_0 =zero at the surface (is now stated in l. 510-511 of the revised manuscript); C_{inf} =0.280 micromolar (Table 2); $D=1.08E-5$ cm^2/s . The choice of $r_1=10$ was because r of 100 or 1000 give a much lower uptake rates than those measured at stagnant conditions for *Chaetoceros*. Completely stagnant conditions are difficult to achieve in the laboratory, because of convection and because cell chains sink. The theoretical calculations were adapted to the units for the actual measurements by SIMS (mole/cell/s). We are grateful to the reviewer that he drew attention to the solution published by Clift et al. 1978. We have exchanged equation 2 (Crank, 1974) with that suggested by the reviewer (Clift et al., 1978). We also tested and compared the calculated rates using the equation by Clift et al. (Eq.1 above) with those calculated using the unpublished solution given by the reviewer (Eq. 2 above) and found less than 1% deviation in the rates calculated by these two different equations. We have now corrected the diffusion coefficients for lower salinity in the brackish water and mentioned those below Eq.2. Thus the diffusion coefficient for nitrate is rather $9.3E-6$. With this assumption Eq. 1 gives a diffusion-limited flux to *Skeletonema* of 26.2 $fmol N cell^{-1} h^{-1}$ and Eq. 2 gives a diffusion-limited flux to *Skeletonema* of 26.4 $fmol N cell^{-1} h^{-1}$. This difference is small although $E=21$ for *Skeletonema*. We therefore exchanged the Equation by Crank with that by Clift et al. in the revised manuscript.*

*This exchange of equations, however, did not alter the conclusions of our study, but rather supported that the theoretical diffusion-limited N flux to *Chaetoceros* (38.4 $fmol N cell^{-1} h^{-1}$) is similar to that measured (37.6 $fmol N cell^{-1} h^{-1}$).*

Lines 39-40—Clarify whether the 24% is of the total or in addition to the total for ambient water. (I learned on p. 9 that it was the latter.)

Author's response: This has been clarified as suggested by the reviewer in the revised manuscript (l.39- 40).

Line 101—Although it is true that viscous dissipation produces heat, it is the shear and not the heat that erodes chemical boundary layers and enhances fluxes. Please clarify.

Author's response: We have clarified this as suggested by the reviewer in the revised manuscript (l. 104)

Line 102—The cited reference (9) does not say that micrometer-sized phytoplankton benefit from shear-produced thinning of their diffusive boundary layers because it is not true. The boundary layer on a cell this small extends only a few micrometers, a scale over which diffusion vastly exceeds advection in delivery speeds. The reference cited hypothesizes that cells 60 μm or more in radius can get substantial increases in arrival fluxes of nutrients from the levels of viscous shear produced by decaying turbulence. Somewhere early on, the authors should state whether they are using equivalent spherical radii or equivalent spherical diameters.

Author's response: We have changed this to "large" phytoplankton (cells $>60 \mu\text{m}$ in radius) (l. 105). The radius or diameter is specified in the various equations used.

Line 262—Some may not consider 5 and 8% to be "considerable." A percentage that small may be exceedingly difficult to detect in the face of natural variability.

Author's response: We have changed "considerable" to "detectable" and elaborated that the effects integrated over a few days may be substantial (l. 284-285).

Lines 440 - 441—Elsewhere (line 454) you say $7.0 \times 10^{-6} \text{ cm}^2 \text{ s}^{-1}$ was used for all the ions. Clarify whether you used different coefficients elsewhere but used only one for all Sherwood number calculations.

Author's response: We used an average diffusion coefficient for nitrate, ammonium, and ortho-phosphate ($\text{HPO}_4^{2-}/\text{H}_2\text{PO}_4^-$) to calculate the Sh because our calculations of diffusion limitations suggest co-limitation by more than one single nutrient. In the revised manuscript we have specified Sh for N and P, respectively (l. 215-217 and l. 278-283).

With a better theoretical context (for diffusion to finite cylinders), I think this manuscript could be a strong contribution.

Sincerely yours,
Peter A. Jumars
Professor Emeritus of Oceanography

Reviewer #2 (Remarks to the Author):

This study seeks to provide single cell data to test the hypothesis that that primary production in chain-forming diatoms is stimulated by turbulence. I was asked to review the application of SIMS in this study, and therefore most of my comments are from that perspective.

I found this a poorly organized and incompletely presented manuscript, and therefore it was very difficult to evaluate, even within my limited scope. The SIMS results are not well organized and are hard to reconcile with each other, particularly on the point of C-assimilation by Chaetoceros.

The figures lack basic reference information, and Fig. 4 in particular is hard to interpret and reconcile with the narrative. The Method Section does not include important information and presents the work as centered on two experiments, which is not how the research is presented in the Results. Finally, the conclusion that turbulence stimulates small-scale biological CO₂ assimilation (L40-41) does not seem to follow from the results: the narrative associated with Fig. 4 states that there is not a turbulence treatment effect for C-assimilation by *Chaetoceros* (L149-151), though the cell chain results make that result hard to understand (L185-187). I did not analyze the N-assimilation results in detail given the substantive problems with the manuscript.

Response: Fig. 4 shows values for the whole populations of diatoms, i.e., all cells in short or long chains. We have made this clearer in the revised manuscript (l. 146-147). A closer analysis showed that C-assimilation is increased by turbulent shear in the cells of the long chains (>7 cells per chain) (Fig. 6) and these long chains also form sinking aggregates due to their large size.

Methodologically, I was concerned that the statistical methods were not discussed in the Method section, particularly because it was not clear how the replicates were handled. I was also concerned about standardization of the measurements, use of the data for the control samples, and measurement precision.

*Response: It is important to remember that we are dealing with natural mixed field-populations of diatoms, and this is one of the innovations in this study. Thus, it is not straightforward to compare bulk measurements of the mixed phytoplankton community with single-cell measurements within one genera. Our analysis showed that the chain-forming diatoms contributed 46% of total nitrate-assimilation in the community but only 12% of total C-assimilation because all phytoplankton assimilate C but not all phytoplankton can assimilate nitrate (Table 3). The cell-to-cell variability in nitrate assimilation rates was much larger (the standard deviation was ca. 50% of the mean value; Fig.5, and Fig. 6) than the variation of the average of nitrate-assimilation at the community level (std was ca. 10% of the mean value; Table 2). We measured a large number of cells to obtain good statistics at the single cell level in the replicates which were closest to the average value of bulk nitrate assimilation rates by the community (GF/F filters analysed by IR-EAMS)(l. 431-433). We analyzed increasing number of cells until the average assimilation rates calculated were stable and representative for the population (l. 242-243 and l. 460-463). In the revised manuscript, we have shown examples of such analysis for *Skeletonema* and *Chaetoceros* (Fig.8) which clearly shows that average values vary largely when the number of cells analysed are below ca. 50, but it stabilizes and reaches representative average values when the number of cells analysed is > 50. We feel confident that our data are indeed representative average values even though the cell-to-cell variation is large. Subsequently, we have used student-T tests, which is valid for comparisons of two populations with a normal distribution and a large number of observations (n). We have reported all relevant numbers of the statistics in brackets throughout the text. The control samples were used to calculate excess isotopic ratios of ¹³C:¹²C and ¹⁵N:¹⁴N. We have also specified this in the text (l. 401-403 and l. 467-468). The high cell-to-cell variation in field populations is important to document in a biological and evolutionary context and is now included in the discussion (l.236-251).*

Finally, based on the SIMS Methods section, the mass resolving power used was not sufficient to resolve what is typically a significant isobaric interference at 27 amu (11B 16O- on ¹³C ¹⁴N-), which would affect the ¹³C/¹²C measurement. This interference is likely the cause of the background in the ¹³C/¹²C images (see below), and therefore the TEP interpretation should be reconsidered.

Response: We agree with the reviewer that part of this signal may be due to a small interference from BO from the pre-sputtered frustules and we have now omitted to mention TEP in this context.

BO- would also be an interference in the measurements on the cells, but it is potentially not significant when the CN- count rate is high. Standard and control measurements are necessary to demonstrate that the cellular $^{13}\text{C}/^{12}\text{C}$ data are valid. These issues will need to be considered in the statistical analysis. If there were other $^{13}\text{C}/^{12}\text{C}$ measurements (e.g., using C dimers or monomers), that could be useful.

Response: We are aware that traces of BO are located in the frustule of diatom cells and may interfere with the $^{13}\text{C}/^{12}\text{C}$ when measurements are not performed in the cell interior, but in or close to the frustule. We have now included the reference by Mejía et al (2013) concerning this issue. The isotopic images (Fig. 2) and the fact that C:N assimilation ratios were close to Redfield demonstrate that our measurements were indeed within the cells where CN-counts are high. Controls (with no added isotopic tracers of ^{13}C -bicarbonate) indeed showed slightly higher $^{13}\text{C}:^{12}\text{C}$ ratios of 0.0112-0.0116 (relative to the natural abundance of 0.0109) which may be due to interference from BO. These control measurements of $^{13}\text{C}/^{12}\text{C}$ in un-amended samples (including a potential signal from BO) were subtracted from the value measured under the ^{13}C -bicarbonate-amended condition to calculate excess $^{13}\text{C}/^{12}\text{C}$ which should be true values of excess $^{13}\text{C}/^{12}\text{C}$ (l. 465-468).

As a general point, based on the method section, there were bulk IRMS data for these experiments, but I could not find them in the manuscript. Given the variability in the SIMS data, bulk C and N assimilation data would be a useful reference for the single cell data.

Response: Please see earlier response. The community (bulk) C and N assimilation data were presented in Table 2 in the original as well as in the revised manuscript. As mentioned earlier, the cell-to-cell variability in nitrate-assimilation rate is much larger (std was ca. 50% of the mean value) than the variation of the average of nitrate-assimilation on a community level (std was ca. 10% of the mean value; Table 2). Thus, variability of IRMS data was low whereas cell-to-cell variability was high. However, our statistical analysis shows that we have analysed enough cells to achieve representative mean values (Fig. 8). We may also mention here that in a N_2 -fixation study during the cyanobacterial summer bloom in the Baltic Sea, we found a 1:1 ratio between IRMS data and SIMS data because N_2 -fixation was confined to a well-defined group of cyanobacteria (Klawonn et al., 2016).

My detailed comments follow:

L66: The description of SIMS does not allow for differences among SIMS instruments. It would be better to be more specific about the critical properties for this study (e.g., imaging, dynamic, magnetic sector).

Response: NanoSIMS and IMS1280 are both dynamic SIMS, with similar magnetic sector mass separation and imaging capabilities but different in spatial resolution (50–100 nm for nanoSIMS compared to ca. 1 μm for IMS1280). The lower spatial resolution of the IMS1280 allows for higher through-put compared to nanoSIMS when cells are large (> 3 micrometer). We have added this information in M&M section (l. 437-439).

L134: The use of the term “stable isotope” in context of the images implies that these images are

different from other secondary ion images, which they are not. I suggest referring to the image data as ion images and ion ratio images or possibly SIMS images.

Response: We have corrected this throughout the manuscript as suggested by the reviewer.

L135: I suggest giving a basic statement on how the image data are used: e.g., isotope ratio data are extracted from individual cells based on cell morphology visualized in the CN ion images.

Response: We have clarified this in the revised manuscript. The cell chains are easy to recognize and distinguish in the $^{12}\text{C}^{14}\text{N}$ ion images (Fig. 2: and l. 455-456). As mentioned below, cell sizes were measured by light microscopy in parallel samples (l. 387-391).

L135-138: The explanation of the difference in relative ^{15}N and ^{13}C enrichment in the TEP is not satisfactory. These ratios reflect the source of the N and C (here, old versus new), not the concentration of each element in the material. Taken on its face, higher ^{13}C enrichment relative to ^{15}N enrichment suggests that the C is new and the N is not. However, based on the reported mass resolving power for the analyses and the species monitored, it is likely that the area around cells in $^{13}\text{C}/^{12}\text{C}$ images reflects a background count rate of $^{11}\text{B}^{16}\text{O}^-$ at 27 amu (see below). This is even more likely since the CN⁻ count rate in the region around the cells is very low (i.e., BO⁻ counts are more likely to be significant). It would be useful to have data for a reference material, such as the substrate around unlabeled diatoms, to rule out this possibility.

Response: It is well-known that Chaetoceros and Skeletonema produce TEP under nutrient limitation (Kjørboe and Hansen, 1983) as also stated in the original submission. However, we agree that Boron, which is a trace nutrient for diatoms, and which is mostly located in the silicified cell walls may be partly responsible for the apparently high counts on the filter due to sputtered material from the silicate frustules. We have rephrased the text accordingly.

If this is an interference issue, it would not necessarily make the data set for the cells (high CN-regions) useless, but the uncertainty on the measurements would have to be re-evaluated.

Response: As mentioned earlier, Boron is a micronutrient and it is mostly located in the silicified cell wall which is pre-sputtered away before the actual measurements are done in the cell interior (with high CN)(Meija et al., 2013). We subtracted the slightly higher $^{13}\text{C}/^{12}\text{C}$ value measured in our control samples when calculating rates. Hence, our data are corrected for this small potential interference by Boron. Cell-specific rates vary largely both in field populations and between different strains of the same species of phytoplankton. SIMS data measured with higher mass resolution (MRP = 11000) which should resolve BO and CN counts on Skeletonema cultures have shown similar cell-specific C-assimilation rates and C:N assimilation ratio as reported in this manuscript when the control measurements were similar to those of the natural abundance of $^{13}\text{C}:^{12}\text{C} = 0.0109$ (own unpubl. data). Hence, we feel confident of our data quality and interpretation.

If the result stands, I would suggest that this discussion not come as the first note on the images since this is an interpretation. I also want to note that I do not think that it is obvious that the images can be directly interpreted for relative enrichment. To the extent that the authors discuss their interpretation of the TEP enrichment, I think it would be better to make the interpretation based on numerical data.

Response: As noticed earlier, we have omitted this comment on the images in the results as suggested by the reviewer. TEP production was never focus of our study.

At the risk of causing confusion, I'll just note here that the way the authors used the word "background" in this context is confusing. The word "background" implies counts that are not from the sample, whereas the authors seem to mean counts that are not directly from the diatoms themselves, but rather from the TEP.

Response: The reviewer is right. We have omitted the word "background" when commenting on the images as suggested by the reviewer.

This, however, is beside the point if BO- is the issue.

Response: please see comment on BO above.

L132: Please be more explicit about the presentation of the nitrate and ammonium experiment data. Based on the Methods section, there are two sets of experiments, but this section does not organize the results in that way, making it more difficult than necessary to relate them to each other.

Response: The nitrate and ammonium experiment data on non-aggregated diatoms are presented in Figure 5. The N-data are discussed relative to the C-assimilation data which were not significantly different in the two experiments (in situ versus laboratory) presumably because light intensity was high enough to saturate photosynthesis and no photoinhibition occurred as intended (l. 151-154).

L143: Are the numbers mean +/- 2 standard errors of the mean?

Response: it is +/- 1 standard error of the mean (l.139).

L147: How do the replicates compare? How are they handled in the statistical test?

Response: Please see earlier explanation.

L167: Is "C-specific C-assimilation" common term?

Response: this term has been used in several NanoSIMS and SIMS studies by now and has the advantage that it is a proxy for growth independent of cell size as we have explained in the text (l. 162-164).

L180-191: It is surprising that there is a significant treatment effect for the *Chaetoceros* cells in the interior of the chains, but not effect overall. Does this mean that most of the cells discussed in the previous section were primarily not in chains? At the start of that section, on L134, it is stated that the data were "measured in single cells within individual cell chains." I am guessing this is just a clarity of writing issue.

*Response: Apical cells have a larger surface area exposed to ambient water. The different responses of *Skeletonema* and *Chaetoceros* may be explained by different morphology and flexibility of the chains (l. 286-289). Diffusion and shear are size-dependent processes (Eq. 2 and Eq.3) as described in the theoretical paragraph (l.84-108). A significant effect of turbulent shear was measured in cell-chains with more than 7 cells (l. 179) and in the figure legend Fig. 6). In fact, this finding is the first of its kind on natural diatom field populations. The data points in Figure 4 represent all measurements performed at a single cell level in chains of various length (≥ 3 cells chain -1). The entire population consists of cells in cell chains. We did not observe a significant effect of shear in the whole diatom*

population but only on the longer chains which also form the sinking aggregates (l. 336-344).

L241: I don't understand. This statement seems to contradict L149-151, which state that there was no statistical difference in cell-specific C-assimilation rates for *Chaetoceros*.

Response: Please see answer above.

L280-282: Unless the authors can produce a high resolution mass spectrum demonstrating that ^{11}B ^{16}O - was resolved from ^{13}C ^{14}N - at mass 27 amu, this line cannot be supported. Also, it would be more convincing if there were significant counts of the species used for the presumed TEP measurements.

Response: We agree that pre-sputtered BO-isotopes (trace nutrient) from the silica frustule may have interfered here and we have omitted the TEP elaboration from the manuscript.

L357: The meaning of "final labeling" is not clear. Does this mean 5% of the bicarbonate in the incubation was labeled? If that is the case, it is relevant to know if this is the calculated abundance of ^{13}C , given that ^{13}C is naturally a 1% isotope.

Response: We have clarified this and explained that the excess $^{13}\text{C}:^{12}\text{C}$ ratio of bicarbonate was 0.036 (l. 395). The uptake rates are calculated on basis of the excess $^{13}\text{C}:^{12}\text{C}$ ratio of bicarbonate in the ambient water, the excess $^{13}\text{C}:^{12}\text{C}$ of the OM, and the incubation time (Eq. 1).

L361-362: Please be more precise. Presumably the authors mean: "and one 1L bottle was incubated with natural, unlabeled NH_4^+ at the same concentration as the other bottles..."

Response: The control bottles were not amended (neither with ^{15}N - nor with ^{14}N -ammonium) to measure the natural isotope ratios of $^{13}\text{C}/^{12}\text{C}$ and $^{15}\text{N}/^{14}\text{N}$ in the diatoms cells as well as in the phytoplankton community. We have clarified this in the text (l. 401-403).

L363-364: If there were nine bottles and two sampling points at which times 3 bottles were sampled, what happened to the last three bottles? Was there a 3rd sampling point? (see question below: was it 2 and 5 hours?) When was the control sampled?

Response: We had three sampling points: T_0 , T_2 and T_5 . The control was sampled at T_5 (but the isotope ratios should be constant at their natural abundance during incubation as this was not amended with isotopic tracers) (l. 404).

L364: 25 h or 2.5 h or something else?

Response: 2 and 5h (l. 404).

L373: The phrasing here raises the question of whether the T_0 samples had been exposed to labels, but immediately sampled, or were not exposed to label. Based on the description above, I understood that it was the latter. If that is the case, "At the start" would be a better way of stating.

Response: yes, it was the latter: "Triplicate incubation bottles without tracer were stopped by filtration at T_0 and after 2 and 5 h" (l. 404).

L385: In the SIMS method section, no mention is made of standardization of the analyses. Please include.

Response: We have run controls on cells which were not exposed to isotopic tracers to compare these with natural abundances. Please see comment above. Considering the standardization procedure of pre-sputtering and measurements please see the following comment below

L391: There is insufficient information to assess the analysis method. What is the basis for stating that silica is removed but the cellular material remains? Do they have an estimate of the depth of sputtering?

Response: The image made without sputtering shows a CN depletion until the Si has been removed. After this the cellular material with high CN is clearly visible as can be seen on the isotope image (Fig. 2) (l. 445-455).

What was the area sputtered with the 10 nA beam?

Response: Slightly larger than the imaged area in order to eliminate possible slight offsets between the two beams (sputter and analytical) and “edge effects” (l. 446-448).

Why “< 5min”?

Response: We calibrated the time needed based on appearance of the cellular material, with a bit of overhead to allow for variations in natural samples – after that we sputtered at a fixed time of 5 min. We have omitted the “<” in the revised manuscript (l. 445).

Why not a more specific sputtering period?

Response: Our test runs showed that we needed 5 min to get through the silica frustule and the interior cell material with high CN content could be imaged. We have clarified this in the revised manuscript (l. 448-450).

Was there a specific marker used to determine that the correct depth was reached?

Response: No, analyses were automated into cells after a fixed period needed to reach the interior of cells with high CN had been assigned (l. 448-450).

Where the cells imaged by SEM after sputtering?

Response: No.

L392: What was the size of the area analyzed? What was the mode of analysis (presumably scanning ion imaging)? If scanning, what was the pixel number and dwell time?

Response: 80x80 μm imaged area, scanning ion imaging, pixels 256x 256. Dwell times: $^{12}\text{C}^{14}\text{N} = 1 \text{ sec.}$, $^{12}\text{C}^{15}\text{N} = 5 \text{ sec.}$, $^{13}\text{C}^{14}\text{N} = 2 \text{ sec.}$; wt. time 0.8 sec, 100 cycles (l. 453-455).

L394: 6000 MRP is not sufficient to resolve $^{13}\text{C}^{14}\text{N}$ - from $^{11}\text{B}^{16}\text{O}$ -, which can interfere at mass 27 amu at relatively high abundance in many samples. The mass difference between $^{13}\text{C}^{14}\text{N}$ - and $^{11}\text{B}^{16}\text{O}$ - is ~2.2 milli-atomic mass units, which requires ~12,000 MRP to resolve, which the CAMECA ims-1280 can do. Boron is a common surface contaminant and it is present in seawater. The

“background” mentioned for Fig. 3 E & F may in fact be BO- counts that have become significant relative to the ^{13}C ^{14}N - counts. Where the CN- count rates are high, BO- may not be significant. Potentially the authors can incorporate this potential source of variability into their statistical evaluation. I am concerned, however, that the number of control samples may limit their statistical power.

*Response: Boron is a trace component in the silica frustule of diatoms (Mejía et al (2013) rather than a surface contaminant. As stated earlier our control samples (analysis in cells which have not been exposed to isotopic tracers) showed slightly higher $^{13}\text{C}:^{12}\text{C}$ than expected from the natural abundance of $^{13}\text{C}:^{12}\text{C}$. However, this number was subtracted from the values recorded with amended samples when calculating excess $^{13}\text{C}:^{12}\text{C}$. We have performed more studies on cultures of *Skeletonema* strains with 11,000 MRP and get similar values of C- and nitrate assimilation rates in the transition to the stationary phase as reported here.*

It would be instructive, but not necessary, to state why the CN isotopomers were used for measuring the $^{13}\text{C}/^{12}\text{C}$ ratio.

Response: They are brighter and have the advantage of allowing two isotope ratios from 3 peaks (l. 452-453).

Did the team also collect C2- for $^{13}\text{C}/^{12}\text{C}$ measurements? That requires lower MRP and could potentially have been collected simultaneously.

Response: No, this was not done.

L400-402: For the C and N assimilation rates, please cite one paper that has the exact procedure used or describe the procedure. Differences among the procedures cannot be reconciled by the reader.

Response: We have now cited Klawonn et al., 2016.

L403: Based on L166, cell size is determined by SIMS images. Was this done using the data processing software?

Response: No, cell sizes were determined in parallel samples by light microscopy as written in the Method section (l. 387-391). Winimage does not allow that as far as we are aware, so ROI's are placed visually.

What were the criteria? 50% height of CN-?

Response: Each image of each cell was carefully examined by eye and the region of interest (ROI) was defined along the border of the $^{12}\text{C}^{14}\text{N}$ image and drawn by hand using the CAMECA software (l. 458-460). The border of the CN image was relatively easy to define (Fig. 2)

Do the authors need to correct for erosion of the lateral dimensions of the cells during high current sputtering?

Response: We did not measure during the high-current sputtering, acquiring data only with the low-current analytical beam. There was no substantial erosion based on the first and last image of the 100 cycles.

L427: The source of the derivation of this equation should be cited, and the limitations should be

provided. Four papers are cited above in the text, which is not sufficiently specific. Note that this equation is inaccurate at high levels of enrichment relative to the pool because of its approximations. The authors should check that their data are inside the accurate range. Popa et al. 2007, ISME J, 1: 354 has a complete derivation of a related equation without approximations.

Response: We are aware of the publication by Popa et al., which is a culture study in which transfer of fixed N₂ from heterocysts to vegetative cells is analyzed in Anabaena by use of nanosims. Only very few cells (< 10 cells) are analyzed in detail per time point. The very extensive analysis of isotopic values (using six equations) presented in that study may be appropriate with a low number cultured cells analyzed and with the aim of studying intercellular processes. This was not the aim of our study in which nutrient transfer between cells does not occur through intercellular transporters as in filamentous cyanobacteria.. The equation we have used is adapted from Montoya et al.(1996) and it has been used in a large number of SIMS publications (in contrast to the approach by Popa et al.) in the (nano) SIMS community . We consider that the equation we used is indeed valid and that values are within the accurate range for field populations within a plankton community study as also confirmed by our other quantitative studies of N₂-fixation in the Baltic Sea where we found a 1:1 ratio between EA-IRMS and SIMS using similar enrichments and approaches because N₂-fixation was confined to a well-defined group of filamentous cyanobacteria (Klawonn et al., 2016). We have used same approach in culture studies and find a very good agreement between growth rates calculated from SIMS measurements and those observed by cell counts over time (pers. unpubl. results).

Fig. 2: This image needs a scale bar.

Response: we have added this to the image

Fig.3. Color scales are hard to read. What are the counts in A and B?

Response: These are counts per pixel. This information has been added to the figure legend.

They are relatively low for CN if that is a cumulative by pixel.

Response: They were sufficiently high for our purposes. As pointed out earlier follow-up experiments on C- and N-assimilation rates and their ratios in Skeletonema have been very reproducible.

How are the ROIs defined?

Response: As mentioned above, the ROI were defined by eye using a freehand polygon tool in Winimage2 based on ¹²C¹⁴N images (l. 458-460).

Fig. 4. I'm having a hard time with this figure. I am not seeing the relative changes indicated in the text. Perhaps these figures should be in the Supplemental, and instead the average and standard deviation of the data should be plotted on a single graph, along with the number of data points represented by each data point.

Response: It is important to document the large cell-to-cell variation in field populations – now included in the discussion (l. 236-251). The average and standard variations are plotted in Fig 5 and Fig. 6.

I find this graph confusing relative to the two experiments called out in the Methods.

Response: Fig. 4 shows the data from the first experiment, whereas Fig. 5 shows the data from both experiments.

Fig.5. It would be helpful to have statistical differences indicated in the graph. Also, it would be helpful to have the number of analyses represented in each bar shown in parentheses to aid reader interpretation.

Response: the requested information is written in the text (line?). We believe that the figure gets overloaded and less clear when number of analyses shown in parentheses are added to the graphs. The number analyses are stated in the figure legend.

Fig.6. “deviation” is misspelled. As above, indicating statistical difference and number of analyses for each bar would be helpful.

Response: We have corrected the spelling accordingly. The requested information was and is written in the text (line?). We believe that the figure gets “too busy” when every information is added to the graphs.

Reviewer #3 (Remarks to the Author):

Review of: Small-scale and large-scale CO₂ sequestration by chain-forming Diatoms are simultaneously stimulated by turbulence in the sea.

By: Johanna Bergkvist, Isabell Klawonn, Martin J. Whitehouse, Gaute Lavik, Volker Brüchert, and Helle Ploug

This paper describes some novel findings that have potentially consequences for our understanding of the oceans' biogeochemistry. It demonstrates, quite convincingly, that the nutrient uptake and growth of large chain-forming diatoms is enhanced under turbulent conditions; conditions under which they also form fast sinking aggregates, thus removing both carbon and nutrients from the surface ocean. The paper is well written, and should be of broad interest to marine and aquatic scientists. I recommend the publication of this manuscript, but draw the authors' attention to a few details that they should consider.

Line 93-95: While I appreciate the intent here, the points made are a bit sloppy. Firstly, I'm not sure what “shorter diffusion distance” refers to. The diffusive flux to a small object is smaller than that to a large object, scaling linearly with size. The second point that is questionable is the reference to “a larger surface area : volume ratio” for small cells. This is true but not germane to the process. Rather it is the linear dimension: cell volume ratio that is pertinent, essentially the ratio of the diffusive flux of nutrients to the demand for growth.

Response: The distance at which concentrations reach 90% of that in the ambient water is a function of the radius (r0) of the organism. Accordingly, the diffusion distance becomes shorter for small organisms relative to large organisms. A larger surface area: volume ratio allows for more nutrient carriers across the cell membrane (which keep concentrations low at the outer surface) relative to the volume which reflects nutrient demands for growth. We have elaborated this in the text (l.94-100).

Line 101: perhaps reword this – it is not the fact that turbulent energy is dissipated as heat that enhances diffusion limited exchange.

Response: The sentence has been changed to: “Below this scale fluid flow is laminar and the turbulent energy is dissipated through viscous shear, which enhances diffusion-limited gas and nutrient exchange in large phytoplankton” (l. 103-105).

Under turbulent conditions as simulated in the in the roller tanks, there are several processes going on at once. There is the increased nutrient flux to individual cells due to shear, there is the coagulation of cells and detrital matter into aggregates with a corresponding increase in sinking speed, and there is the bacterial colonization of these aggregates with subsequent remineralization of detrital nutrients. Each one of these processes has an impact on the nutrient dynamics and growth of the diatoms. It would be really helpful for the reader if the authors could try to dis-entangle these processes. The central premise, surmised from Figure 1, is that increased uptake due to turbulent shear is the candidate mechanism, but can the other processes be discounted. At the very least, this should be mentioned in the discussion.

Response: We agree with the reviewer, and we have included a paragraph discussing aggregation and degradation processes and their time constraints compared to sinking velocities (l. 336-355). We measured ^{13}C -, $^{15}\text{NO}_3$ - and $^{15}\text{NH}_4^+$ assimilation in non-aggregated diatom chains (l. 444) during short term incubations: 2h, and 5 h for NH_4^+ (l. 404) and 8.5 h for nitrate (l. 408) and $^{14}\text{NH}_4^+$ production in diatom aggregates (12h incubation) after their formation in roller tanks during 24 h (l. 496-498). We did not measure growth in aggregated diatoms (it is not possible with SIMS when chains are on top of each other). We have added more information on the low cell-specific NH_4^+ assimilation in non-aggregated diatoms during darkness (l. 325-329). The high net NH_4^+ fluxes from aggregates and the high concentration inside diatom aggregates are indicative of net degradation rather than net assimilation in the aggregate as microbial community. Hence, we did not consider the growth of diatoms in the aggregates – rather their degradation. (l. 349-352).

On the behalf of all authors,

Helle Ploug

Reviewers' comments:

Reviewer #1 (Remarks to the Author):

The authors have done a conscientious job of responding to my suggestions. They have made it much easier to follow their methods and calculations. I look forward to seeing this paper and the further research it will stimulate appear in print.

I found a couple of minor issues for author attention:

In the first paragraph of methods, salinity should be reported as "Practical salinity, S_p , was 6." See <http://www.teos-10.org> for current SI practice. The text reports a Couette shear rate of 20 s^{-1} , whereas the methods report 21 s^{-1} .

Reviewer #2, Round two, General comments: I thank for the authors for their detailed responses to my review. This study to test the long-standing hypothesis that turbulence stimulates primary production is obviously of strong potential interest considering that it has been sent out for a second round of review despite my initial review. In deference to everyone involved, I have more than doubled my time evaluating this manuscript, both to reconsider my initial evaluation and consider the responses to my initial review. Unfortunately, I cannot give a significantly more positive review, and in fact, the responses to my review comments have provided me with a better understanding of the flaws in the data. The most fundamental problem is that the replication in the experiment was not used for evaluating the central hypothesis, but instead it seems that all data for each treatment were averaged together. Furthermore, the statistical methods used are insufficient to account for the multiple comparisons that are made (i.e., single cells versus chains of 2 cells versus chains of 3 cells, up to >7 cells per chain) and to account for the likelihood of non-normal data distributions for subsets of the total data set.

The manuscript and characterization of the results are also problematic. The Abstract does not present the nuanced nature of the data, particularly for *Chaetoceros*, or the primary metrics that the authors are asserting are the basis for support of the central hypothesis (“up to 59%” is not representative). The paper does not provide basic information, such as the bulk rate of fixation for the replicates under the different treatments, how C-assimilation differed by chain length, and the contribution of longer chain lengths to overall fixation. The absence of this information makes it impossible for the reader put the presented results in context. Basic questions I have include: Is there enough data by chain length to make an evaluation? Is there a trend? What is the justification for making >7 cell chains a subgroup? What are the subgroups’ contributions to total C assimilation? Are longer chains more productive? (I also wonder if the data suggest other related or unrelated hypotheses that others might want to pursue, such as, do the data provide information related to cell division?) I also find it unnecessarily difficult to determine in which cases there are statistically significant effects in Figures 4, 5 & 6.

Overall, I am concerned that the data as they stand present weak and poorly characterized support for the central hypothesis. From my perspective, when one is claiming to demonstrate a paradigm, the experimental design and execution and the data presentation and evaluation need to be rigorous, and the results need to be unambiguous. Unfortunately, for the reasons laid out above, this work does not meet this standard. However, I would consider this study publishable even if the results are ambiguous, if the data were more rigorously handled and the results were described in the Abstract in a balanced way.

For lack of a better place to note this, in Lines 141-143 I think there is a mistake here, because in Figure 5, it does not look to me that *Skeletonema* had 32% higher C assimilation under turbulent conditions. Is this supposed to be “C:N assimilation”? That would make more sense since this statement is associated with the C:N discussion above. Whatever the answer, this would good information for the Abstract.

My responses are below.

Reviewer #2 (Remarks to the Author): This study seeks to provide single cell data to test the hypothesis that that primary production in chainforming diatoms is stimulated by turbulence. I was asked to review the application of SIMS in this study, and therefore most of my comments are from that perspective.

I found this a poorly organized and incompletely presented manuscript, and therefore it was very difficult to evaluate, even within my limited scope. The SIMS results are not well organized and are hard to

reconcile with each other, particularly on the point of C-assimilation by Chaetoceros. The figures lack basic reference information, and Fig. 4 in particular is hard to interpret and reconcile with the narrative. The Method Section does not include important information and presents the work as centered on two experiments, which is not how the research is presented in the Results. Finally, the conclusion that turbulence stimulates small-scale biological CO₂ assimilation (L40-41) does not seem to follow from the results: the narrative associated with Fig. 4 states that there is not a turbulence treatment effect for C-assimilation by Chaetoceros (L149-151), though the cell chain results make that result hard to understand (L185-187). I did not analyze the N-assimilation results in detail given the substantive problems with the manuscript.

Response: Fig. 4 shows values for the whole populations of diatoms, i.e., all cells in short or long chains. We have made this clearer in the revised manuscript (l. 146-147). A closer analysis showed that C-assimilation is increased by turbulent shear in the cells of the long chains (>7 cells per chain) (Fig. 6) and these long chains also form sinking aggregates due to their large size.

Reviewer Response: The statistical justification for separating out chains of >7 cells is not presented, the data are not handled based on the replicates, the statistics used are not justified for these small sample numbers, and the quantitative importance of the observed effect is not presented.

Methodologically, I was concerned that the statistical methods were not discussed in the Method section, particularly because it was not clear how the replicates were handled. I was also concerned about standardization of the measurements, use of the data for the control samples, and measurement precision.

Response: It is important to remember that we are dealing with natural mixed field-populations of diatoms, and this is one of the innovations in this study. Thus, it is not straightforward to compare bulk measurements of the mixed phytoplankton community with single-cell measurements within one genera. Our analysis showed that the chain-forming diatoms contributed 46% of total nitrate assimilation in the community but only 12% of total C-assimilation because all phytoplankton assimilate C but not all phytoplankton can assimilate nitrate (Table 3). The cell-to-cell variability in nitrate assimilation rates was much larger (the standard deviation was ca. 50% of the mean value; Fig.5, and Fig. 6) than the variation of the average of nitrate-assimilation at the community level (std was ca. 10% of the mean value; Table 2). We measured a large number of cells to obtain good statistics at the single cell level in the replicates which were closest to the average value of bulk nitrate assimilation rates by the community (GF/F filters analysed by IR-EAMS)(l. 431-433). We analyzed increasing number of cells until the average assimilation rates calculated were stable and representative for the population (l. 242-243 and l. 460-463). In the revised manuscript, we have shown examples of such analysis for Skeletonema and Chaetoceros (Fig.8) which clearly shows that average values vary largely when the number of cells analysed are below ca. 50, but it stabilizes and reaches representative average values when the number of cells analysed is > 50. We feel confident that our data are indeed representative average values even though the cell-to-cell variation is large. Subsequently, we have used student-T tests, which is valid for comparisons of two populations with a normal distribution and a large number of observations (n). We have reported all relevant numbers of the statistics in brackets throughout the text. The control samples were used to calculate excess isotopic ratios of ¹³C:¹²C and ¹⁵N:¹⁴N. We have also specified this in the text (l. 401-403 and l. 467- 468). The high cell-to-cell variation in field populations is important to document in a biological and evolutionary context and is now included in the discussion (l.236-251).

Reviewer Response: The Method section still lacks details on the use of the replicates, the statistical tests and the use of the control samples and standardization. Based on the statement above, the authors combined the data from the replicates, which reduces the experiment to an n of one, which is not a good basis for claiming to demonstrate a paradigm. The authors should re-analyze the data, separating out the replicates, and which should then be statistically analyzed, taking into account whether the data are normally distributed and the fact that there are multiple comparisons.

Finally, based on the SIMS Methods section, the mass resolving power used was not sufficient to resolve what is typically a significant isobaric interference at 27 amu (11B 16O- on 13C 14N-), which would affect the 13C/12C measurement. This interference is likely the cause of the background in the 13C/12C images (see below), and therefore the TEP interpretation should be reconsidered.

Response: We agree with the reviewer that part of this signal may be due to a small interference from BO from the pre-sputtered frustules and we have now omitted to mention TEP in this context.

Okay.

BO- would also be an interference in the measurements on the cells, but it is potentially not significant when the CN- count rate is high. Standard and control measurements are necessary to demonstrate that the cellular 13C/12C data are valid. These issues will need to be considered in the statistical analysis. If there were other 13C/12C measurements (e.g., using C dimers or monomers), that could be useful.

Response: We are aware that traces of BO are located in the frustule of diatom cells and may interfere with the 13C/12C when measurements are not performed in the cell interior, but in or close to the frustule. We have now included the reference by Mejía et al (2013) concerning this issue. The isotopic images (Fig. 2) and the fact that C:N assimilation ratios were close to Redfield demonstrate that our measurements were indeed within the cells where CN-counts are high. Controls (with no added isotopic tracers of 13C-bicarbonate) indeed showed slightly higher 13C:12C ratios of 0.0112-0.0116 (relative to the natural abundance of 0.0109) which may be due to interference from BO. These control measurements of 13C/12C in un-amended samples (including a potential signal from BO) were subtracted from the value measured under the 13C-bicarbonate-amended condition to calculate excess 13C/12C which should be true values of excess 13C/12C (l. 465-468).

Reviewer Response: While I agree that the data appear valid, I think it is necessary that the flaws in the instrument set up be made explicit and up front in the Methods section. I do not want this paper to become a reference for how to make these measurements, especially if published in a Nature journal. It has long been established that boron is a ubiquitous surface contaminant that produces BO- and can interfere with CN- measurements at mass 27 (Zinner et al., 1989, Geochimica et Cosmochimica Acta 53, 3273–3290). Furthermore, 6000 MRP is only marginally high enough to resolve 13C2- from 12C14N-, which is unimportant in natural samples, but which can become significant in 13C-labeled samples or low nitrogen samples (Zinner et al., 1989 is a reasonable reference. Unfortunately, Peteranderl and Lechene (J Am Soc Mass Spectrom 2004, 15, 478–485) mislabel the 13C2 peak, though they correctly present the required MRP to be ~7000). Therefore, the text in the Methods section should clearly acknowledge that the mass resolving power used was in error—concurrent with the explanation for why these species were used. Starting from the text on line 465, I propose something like:

<<For each cell chain we recorded secondary ion (SIMS) images of $^{13}\text{C}^{14}\text{N}^-$ and $^{12}\text{C}^{14}\text{N}^-$, and $^{12}\text{C}^{15}\text{N}^-$ using a peak-switching routine at a mass resolution of ca. 6000 ($M/\Delta M$), which is less than the $\sim 12,000$ $M/\Delta M$ necessary to resolve BO^- from $^{13}\text{C}^{14}\text{N}^-$ (Zinner et al., 1989); BO^- is a common surface contaminant and present in the diatom frustules [ref]. Also, ~ 7000 $M/\Delta M$ is necessary to fully resolve $^{12}\text{C}^{14}\text{N}^-$ from $^{13}\text{C}^{13}\text{C}^-$, which can be significant for highly ^{13}C -enriched samples. Measurements on control samples suggest that these interferences did not significantly affect the isotope measurements on the cells, but BO^- did affect ratios from low yield regions. The use of CN isotopomers have the advantage that they are bright and two isotope ratios can be measured from measurements of three isotopomers.>>

Given the analytical issues, the standard and control data should at least be presented in a supplemental section, and the uncorrected statistics (mean, standard deviation and n) should at least be presented in the Methods section, and the range of measured ^{13}C enrichment should be reported.

As a general point, based on the method section, there were bulk IRMS data for these experiments, but I could not find them in the manuscript. Given the variability in the SIMS data, bulk C and N assimilation data would be a useful reference for the single cell data.

Response: Please see earlier response. The community (bulk) C and N assimilation data were presented in Table 2 in the original as well as in the revised manuscript. As mentioned earlier, the cell-to-cell variability in nitrate-assimilation rate is much larger (std was ca. 50% of the mean value) than the variation of the average of nitrate-assimilation on a community level (std was ca. 10% of the mean value; Table 2). Thus, variability of IRMS data was low whereas cell-to-cell variability was high. However, our statistical analysis shows that we have analysed enough cells to achieve representative mean values (Fig. 8). We may also mention here that in a N_2 -fixation study during the cyanobacterial summer bloom in the Baltic Sea, we found a 1:1 ratio between IRMS data and SIMS data because N_2 -fixation was confined to a well-defined group of cyanobacteria (Klawonn et al., 2016).

Reviewer Response: This is helpful because I did not understand how the IRMS data were used, but I am still confused by why the results for “still” and “turbulent” treatment data aren’t compared. Based on the author response, the treatment effect should be detected in the bulk data, and if this is true, it would be a substantial piece of evidence showing the significance of the hypothesized effect. However, if the effect is not measurable in the bulk data, it should still be presented in Table 2. Also, uncertainties should be calculated for C:N and C:other ratios using Gaussian error propagation and presented in Table 2.

My detailed comments follow:

L66: The description of SIMS does not allow for differences among SIMS instruments. It would be better to be more specific about the critical properties for this study (e.g., imaging, dynamic, magnetic sector).

Response: NanoSIMS and IMS1280 are both dynamic SIMS, with similar magnetic sector mass separation and imaging capabilities but different in spatial resolution (50-100 nm for nanoSIMS

compared to ca. 1 μm for IMS1280). The lower spatial resolution of the IMS1280 allows for higher through-put compared to nanoSIMS when cells are large (> 3 micrometer). We have added this information in M&M section (l. 437-439).

Response of reviewer: I would suggest taking a step back on the instrument explanation, given that TOF-SIMS instruments are more common than dynamic SIMS instruments and this manuscript is aimed at a broad audience. A key point of the instrument used is that it has high mass resolving power with flat-top peaks, which enables high precision and accuracy isotope measurements.

The argument that these analyses are faster than nanoSIMS, if true, would be interesting, but the authors have not substantiated this claim, and the method description is missing key information. Lower spatial resolution, per se, does not allow for higher throughput, and higher beam current (which the authors seem to be associating with lower spatial) does not necessarily allow for higher throughput either. The analysis currents used in this study can easily be achieved with a NanoSIMS 50L. For these analyses to be substantially faster, the difference would have to be the detector, which is not specified in the manuscript. Typically, imaging on any dynamic SIMS instrument is performed with electron multipliers because Faraday cups are too slow. In turn, it is the electron multipliers that tend to limit the speed of the analysis because analyst limit the secondary ion count rate because high count rates result in premature detector failure. It would be very interesting if the detector used was an EM that allows for high (>1 000 000 counts per second) secondary ion currents or if the analyses was somehow performed with a Faraday cup (though this would not likely be faster because of the slow Faraday cup response). However, if the analyses were performed with standard electron multipliers, the analysis time would be similar for the IMS 1280 and a NanoSIMS.

In any case, the authors should specify the detector used and the secondary ion currents obtained since that is the critical part of the analysis speed.

On the topic of speed, there is a problem with the dwell time given in the Methods (L454: “The dwell times were 1 s, 5 s and 2 s....”). My guess is that the units should be milliseconds, not seconds, as seconds would yield very long analyses. Also, please write “dwell time per pixel” and “ms/pixel” for clarity sake.

L134: The use of the term “stable isotope” in context of the images implies that these images are different from other secondary ion images, which they are not. I suggest referring to the image data as ion images and ion ratio images or possibly SIMS images.

Response: We have corrected this throughout the manuscript as suggested by the reviewer.

Okay

L135: I suggest giving a basic statement on how the image data are used: e.g., isotope ratio data are extracted from individual cells based on cell morphology visualized in the CN ion images.

Response: We have clarified this in the revised manuscript. The cell chains are easy to recognize and distinguish in the $^{12}\text{C}^{14}\text{N}$ ion images (Fig. 2: and l. 455-456). As mentioned below, cell sizes were measured by light microscopy in parallel samples (l. 387-391).

Okay

L135-138: The explanation of the difference in relative ^{15}N and ^{13}C enrichment in the TEP is not satisfactory. These ratios reflect the source of the N and C (here, old versus new), not the concentration of each element in the material. Taken on its face, higher ^{13}C enrichment relative to ^{15}N enrichment suggests that the C is new and the N is not. However, based on the reported mass resolving power for the analyses and the species monitored, it is likely that the area around cells in $^{13}\text{C}/^{12}\text{C}$ images reflects a background count rate of $^{11}\text{B}^{16}\text{O}^-$ at 27 amu (see below). This is even more likely since the CN- count rate in the region around the cells is very low (i.e., BO^- counts are more likely to be significant). It would be useful to have data for a reference material, such as the substrate around unlabeled diatoms, to rule out this possibility.

Response: It is well-known that Chaetoceros and Skeletonema produce TEP under nutrient limitation (Kjørboe and Hansen, 1983) as also stated in the original submission. However, we agree that Boron, which is a trace nutrient for diatoms, and which is mostly located in the silicified cell walls may be partly responsible for the apparently high counts on the filter due to sputtered material from the silicate frustules. We have rephrased the text accordingly.

Okay

If this is an interference issue, it would not necessarily make the data set for the cells (high CN regions) useless, but the uncertainty on the measurements would have to be re-evaluated.

Response: As mentioned earlier, Boron is a micronutrient and it is mostly located in the silicified cell wall which is pre-sputtered away before the actual measurements are done in the cell interior (with high CN) (Meija et al., 2013). We subtracted the slightly higher $^{13}\text{C}/^{12}\text{C}$ value measured in our control samples when calculating rates. Hence, our data are corrected for this small potential interference by Boron. Cell-specific rates vary largely both in field populations and between different strains of the same species of phytoplankton. SIMS data measured with higher mass resolution (MRP = 11000) which should resolve BO^- and CN counts on Skeletonema cultures have shown similar cell-specific

C-assimilation rates and C:N assimilation ratio as reported in this manuscript when the control measurements were similar to those of the natural abundance of $^{13}\text{C}:^{12}\text{C} = 0.0109$ (own unpubl. data). Hence, we feel confident of our data quality and interpretation.

Reviewer response: If I understand this correctly, the SIMS data for the one control sample were used to calculate the single cell assimilation rates using equation 1. The data for these should be reported in the paper, presumably in the SIMS method section.

If the result stands, I would suggest that this discussion not come as the first note on the images since this is an interpretation. I also want to note that I do not think that it is obvious that the images can be directly interpreted for relative enrichment. To the extent that the authors discuss their interpretation of the TEP enrichment, I think it would be better to make the interpretation based on numerical data.

Response: As noticed earlier, we have omitted this comment on the images in the results as suggested by the reviewer. TEP production was never focus of our study.

Okay

At the risk of causing confusion, I'll just note here that the way the authors used the word "background" in this context is confusing. The word "background" implies counts that are not from the sample, whereas the authors seem to mean counts that are not directly from the diatoms themselves, but rather from the TEP.

Response: The reviewer is right. We have omitted the word "background" when commenting on the images as suggested by the reviewer.

Okay

This, however, is beside the point if BO- is the issue.

Response: please see comment on BO above.

Okay

L132: Please be more explicit about the presentation of the nitrate and ammonium experiment data. Based on the Methods section, there are two sets of experiments, but this section does not organize the results in that way, making it more difficult than necessary to relate them to each other.

Response: The nitrate and ammonium experiment data on non-aggregated diatoms are presented in Figure 5. The N-data are discussed relative to the C-assimilation data which were not significantly different in the two experiments (in situ versus laboratory) presumably because light intensity was high enough to saturate photosynthesis and no photoinhibition occurred as intended (l. 151-154).

Reviewer Response: My comment here was at a more general level. I found the nitrogen assimilation data poorly integrated in general. I would even refer it back to the Abstract, in which the paper is framed in terms of primary production, and nitrogen assimilation seems tacked on.

L143: Are the numbers mean +/- 2 standard errors of the mean?

Response: it is +/- 1 standard error of the mean (l.139).

Okay

L147: How do the replicates compare? How are they handled in the statistical test?

Response: Please see earlier explanation.

Reviewer Response: As noted above, it is normal and expected in ecology for there to be replication to control for natural variability, which is often high. In the Method section, the authors state that they performed replication (L398-400), but I cannot find any presentation of how the replication is used. My impression from the paper and the Author Response is that they combine the data from all of the replicates together, which would make the results essentially anecdotal. To provide a more robust test of the central hypothesis, the treatments should have replication.

L167: Is “C-specific C-assimilation” common term?

Response: this term has been used in several NanoSIMS and SIMS studies by now and has the advantage that it is a proxy for growth independent of cell size as we have explained in the text (l. 162-164).

Okay

L180-191: It is surprising that there is a significant treatment effect for the *Chaetoceros* cells in the interior of the chains, but not effect overall. Does this mean that most of the cells discussed in the previous section were primarily not in chains? At the start of that section, on L134, it is stated that the data were “measured in single cells within individual cell chains.” I am guessing this is just a clarity of writing issue.

*Response: Apical cells have a larger surface area exposed to ambient water. The different responses of *Skeletonema* and *Chaetoceros* may be explained by different morphology and flexibility of the chains (l. 286-289). Diffusion and shear are size-dependent processes (Eq. 2 and Eq.3) as described in the theoretical paragraph (l.84-108). A significant effect of turbulent shear was measured in cell chains with more than 7 cells (l. 179) and in the figure legend Fig. 6). In fact, this finding is the first of its kind on natural diatom field populations. The data points in Figure 4 represent all measurements performed at a single cell level in chains of various length (≥ 3 cells chain⁻¹). The entire population consists of cells in cell chains. We did not observe a significant effect of shear in the whole diatom population but only on the longer chains which also form the sinking aggregates (l. 336-344).*

Reviewer Response: This explanation of the results makes sense to me. If this result holds up under a more rigorous treatment of the data, I would suggest putting this level of subtlety into the Abstract to set up the paper. As the Abstract stands, these results seem inconsistent with the broad brush claim of demonstrating that turbulent shear increases primary productivity.

L241: I don't understand. This statement seems to contradict L149-151, which state that there was no statistical difference in cell-specific C-assimilation rates for *Chaetoceros*.

Response: Please see answer above.

Okay.

L280-282: Unless the authors can produce a high resolution mass spectrum demonstrating that ^{11}B ^{16}O - was resolved from ^{13}C ^{14}N - at mass 27 amu, this line cannot be supported. Also, it would be more convincing if there were significant counts of the species used for the presumed TEP measurements.

Response: We agree that pre-sputtered BO-isotopes (trace nutrient) from the silica frustule may have interfered here and we have omitted the TEP elaboration from the manuscript.

Okay

L357: The meaning of “final labeling” is not clear. Does this mean 5% of the bicarbonate in the incubation was labeled? If that is the case, it is relevant to know if this is the calculated abundance of ^{13}C , given that ^{13}C is naturally a 1% isotope.

Response: We have clarified this and explained that the excess $^{13}\text{C}:^{12}\text{C}$ ratio of bicarbonate was 0.036 (l. 395). The uptake rates are calculated on basis of the excess $^{13}\text{C}:^{12}\text{C}$ ratio of bicarbonate in the ambient water, the excess $^{13}\text{C}:^{12}\text{C}$ of the OM, and the incubation time (Eq. 1).

Okay

L361-362: Please be more precise. Presumably the authors mean: “and one 1L bottle was incubated with natural, unlabeled NH_4^+ at the same concentration as the other bottles...”

Response: The control bottles were not amended (neither with ^{15}N - nor with ^{14}N -ammonium) to measure the natural isotope ratios of $^{13}\text{C}/^{12}\text{C}$ and $^{15}\text{N}/^{14}\text{N}$ in the diatoms cells as well as in the phytoplankton community. We have clarified this in the text (l. 401-403).

Okay

L363-364: If there were nine bottles and two sampling points at which times 3 bottles were sampled, what happened to the last three bottles? Was there a 3rd sampling point? (see question below: was it 2 and 5 hours?) When was the control sampled?

Response: We had three sampling points: T_0 , T_2 and T_5 . The control was sampled at T_5 (but the isotope ratios should be constant at their natural abundance during incubation as this was not amended with isotopic tracers) (l. 404).

Reviewer Response: I am sorry if I missed this explanation in the initial manuscript, but as you might imagine, this leaves me with the question, How are these time points used in the manuscript? I do not see any indication of time in any of the graphs. Showing a temporal trend would support the central hypothesis. Please include some explanation in the manuscript or remove mention of the time points if not used.

L364: 25 h or 2.5 h or something else?

Response: 2 and 5h (l. 404).

Okay.

L373: The phrasing here raises the question of whether the T0 samples had been exposed to labels, but immediately sampled, or were not exposed to label. Based on the description above, I understood that it was the latter. If that is the case, “At the start” would be a better way of stating.

Response: yes, it was the latter: “Triplicate incubation bottles without tracer were stopped by filtration at T₀ and after 2 and 5 h” (l. 404).

Okay.

L385: In the SIMS method section, no mention is made of standardization of the analyses. Please include.

Response: We have run controls on cells which were not exposed to isotopic tracers to compare these with natural abundances. Please see comment above. Considering the standardization procedure of pre-sputtering and measurements please see the following comment below

Reviewer Response: If I am not missing it, the T0 data should be reported separately from the unlabeled control.

L391: There is insufficient information to assess the analysis method. What is the basis for stating that silica is removed but the cellular material remains? Do they have an estimate of the depth of sputtering?

Response: The image made without sputtering shows a CN depletion until the Si has been removed. After this the cellular material with high CN is clearly visible as can be seen on the isotope image (Fig. 2) (l. 445-455).

Reviewer Response: This would be good information to include in the Method section.

What was the area sputtered with the 10 nA beam?

Response: Slightly larger than the imaged area in order to eliminate possible slight offsets between the two beams (sputter and analytical) and “edge effects” (l. 446-448).

Reviewer Response: The information above is very helpful, but still it is better to be more precise, if only giving a range.

Why “< 5min”?

Response: We calibrated the time needed based on appearance of the cellular material, with a bit of overhead to allow for variations in natural samples - after that we sputtered at a fixed time of 5 min. We have omitted the “<” in the revised manuscript (l. 445).

Okay.

Why not a more specific sputtering period?

Response: Our test runs showed that we needed 5 min to get through the silica frustule and the interior cell material with high CN content could be imaged. We have clarified this in the revised manuscript (l. 448-450).

Okay

Was there a specific marker used to determine that the correct depth was reached?

Response: No, analyses were automated into cells after a fixed period needed to reach the interior of cells with high CN had been assigned (l. 448-450).

Okay

Where the cells imaged by SEM after sputtering?

Response: No.

Okay

L392: What was the size of the area analyzed? What was the mode of analysis (presumably scanning ion imaging)? If scanning, what was the pixel number and dwell time?

Response: 80x80 μm imaged area, scanning ion imaging, pixels 256x 256. Dwell times: 12C14N = 1 sec., 12C15N = 5 sec., 13C14N = 2 sec.; wt. time 0.8 sec, 100 cycles (l. 453-455).

Reviewer Response: Thank you for adding this information. As mentioned above; I am guessing it should be “milliseconds per pixel”.

L394: 6000 MRP is not sufficient to resolve 13C 14N- from 11B 16O-, which can interfere at mass 27 amu at relatively high abundance in many samples. The mass difference between 13C 14N- and 11B 16O- is ~2.2 milli-atomic mass units, which requires ~12,000 MRP to resolve, which the CAMECA ims-1280 can do. Boron is a common surface contaminant and it is present in seawater. The “background” mentioned for Fig. 3 E & F may in fact be BO- counts that have become significant relative to the 13C 14N- counts. Where the CN- count rates are high, BO- may not be significant. Potentially the authors can incorporate this potential source of variability into their statistical evaluation. I am concerned, however, that the number of control samples may limit their statistical power.

Response: Boron is a trace component in the silica frustule of diatoms (Mejía et al (2013) rather than a surface contaminant. As stated earlier our control samples (analysis in cells which have not been exposed to isotopic tracers) showed slightly higher $^{13}\text{C}:^{12}\text{C}$ than expected from the natural abundance of $^{13}\text{C}:^{12}\text{C}$. However, this number was subtracted from the values recorded with amended samples when calculating excess $^{13}\text{C}:^{12}\text{C}$. We have performed more studies on cultures of Skeletonema strains with 11,000 MRP and get similar values of C- and nitrate assimilation rates in the transition to the stationary phase as reported here.

Reviewer Response: In fact, boron is a common contaminant. The fact that it is in the frustule only makes the concern great, though I support the conclusion that the data are okay.

It would be instructive, but not necessary, to state why the CN isotopomers were used for measuring the $^{13}\text{C}/^{12}\text{C}$ ratio.

Response: They are brighter and have the advantage of allowing two isotope ratios from 3 peaks (l. 452-453).

Thank you for adding that.

Did the team also collect C2- for $^{13}\text{C}/^{12}\text{C}$ measurements? That requires lower MRP and could potentially have been collected simultaneously.

Response: No, this was not done.

Okay

L400-402: For the C and N assimilation rates, please cite one paper that has the exact procedure used or describe the procedure. Differences among the procedures cannot be reconciled by the reader.

Response: We have now cited Klawonn et al., 2016.

Okay

L403: Based on L166, cell size is determined by SIMS images. Was this done using the data processing software?

Response: No, cell sizes were determined in parallel samples by light microscopy as written in the Method section (l. 387-391). Winimage does not allow that as far as we are aware, so ROI's are placed visually.

Thank you for that clarification.

What were the criteria? 50% height of CN-?

Response: Each image of each cell was carefully examined by eye and the region of interest (ROI) was defined along the border of the $^{12}\text{C}^{14}\text{N}$ image and drawn by hand using the CAMECA software (l. 458-460). The border of the CN image was relatively easy define (Fig. 2)

Okay

Do the authors need to correct for erosion of the lateral dimensions of the cells during high current sputtering?

Response: We did not measure during the high-current sputtering, acquiring data only with the low current analytical beam. There was no substantial erosion based on the first and last image of the 100 cycles.

Reviewer Response: This is taken care of. I meant for the cell size, but now I understand that cell size was based parallel samples.

L427: The source of the derivation of this equation should be cited, and the limitations should be provided. Four papers are cited above in the text, which is not sufficiently specific. Note that this equation is inaccurate at high levels of enrichment relative to the pool because of its approximations. The authors should check that their data are inside the accurate range. Popa et al. 2007, ISME J, 1: 354 has a complete derivation of a related equation without approximations.

*Response: We are aware of the publication by Popa et al., which is a culture study in which transfer of fixed N_2 from heterocysts to vegetative cells is analyzed in *Anabaena* by use of nanosims. Only very few cells (< 10 cells) are analyzed in detail per time point. The very extensive analysis of isotopic values (using six equations) presented in that study may be appropriate with a low number cultured cells analyzed and with the aim of studying intercellular processes. This was not the aim of our study in which nutrient transfer between cells does not occur through intercellular transporters as in filamentous cyanobacteria.. The equation we have used is adapted from Montoya et al.(1996) and it has been used in a large number of SIMS publications (in contrast to the approach by Popa et al.) in the (nano) SIMS community . We consider that the equation we used is indeed valid and that values are within the accurate range for field populations within a plankton community study as also confirmed by our other quantitative studies of N_2 -fixation in the Baltic Sea where we found a 1:1 ratio between EA-IRMS and SIMS using similar enrichments and approaches because N_2 -fixation was confined to a well-defined group of filamentous cyanobacteria (Klawonn et al., 2016). We have used same approach in culture studies and find a very good agreement between growth rates calculated from SIMS measurements and those observed by cell counts over time (pers. unpubl. results).*

Reviewer Response: Thank you for this information. I am confused by why Montoya et al is not directly cited with the equation. My recollection of Adam et al is that it did not link Montoya et al to this equation, so if I am correct, I would recommend against using it and only citing one paper—presumably Klawonn et al., 2016 if it cites Montoya et al.—so that the reader does not have to look through three papers to find the full source.

Fig. 2: This image needs a scale bar.

Response: we have added this to the image

Okay.

Fig.3. Color scales are hard to read. What are the counts in A and B?

Response: These are counts per pixel. This information has been added to the figure legend.

Okay. This addresses my concern about possibly low total counts below.

They are relatively low for CN if that is a cumulative by pixel.

Response: They were sufficiently high for our purposes. As pointed out earlier follow-up experiments on C- and N-assimilation rates and their ratios in Skeletonema have been very reproducible.

Reviewer Response: Okay. The clarification in the previous response explains why the numbers can be low but still sufficient.

How are the ROIs defined?

Response: As mentioned above, the ROI were defined by eye using a freehand polygon tool in Winimage2 based on 12C14N images (l. 458-460).

Okay.

Fig. 4. I'm having a hard time with this figure. I am not seeing the relative changes indicated in the text. Perhaps these figures should be in the Supplemental, and instead the average and standard deviation of the data should be plotted on a single graph, along with the number of data points represented by each data point.

Response: It is important to document the large cell-to-cell variation in field populations - now included in the discussion (l. 236-251). The average and standard variations are plotted in Fig 5 and Fig. 6.

Reviewer Response: While I appreciate showing the variability, I do not think it helps to make the authors' case.

I find this graph confusing relative to the two experiments called out in the Methods.

Response: Fig. 4 shows the data from the first experiment, whereas Fig. 5 shows the data from both experiments.

Reviewer Response: It would be helpful to the reader to put this information in the captions.

Fig.5. It would be helpful to have statistical differences indicated in the graph. Also, it would be helpful to have the number of analyses represented in each bar shown in parentheses to aid reader interpretation.

Response: the requested information is written in the text (line?). We believe that the figure gets overloaded and less clear when number of analyses shown in parentheses are added to the graphs. The number analyses are stated in the figure legend.

Reviewer Response: I suppose this is an editorial decision. Busy or not, I think this presentation undermines the authors' case, particularly in Figure 6, where it doesn't look as if there are significant differences and it is extra work to try to figure out which comparisons are statistically significant. At least using standard error of the mean would help, or perhaps a non-parametric variance if for small sample numbers.

Note omission of "(D)" in Figure 5 caption, L771.

Fig.6. "deviation" is misspelled. As above, indicating statistical difference and number of analyses for each bar would be helpful.

Response: We have corrected the spelling accordingly. The requested information was and is written in the text (line?). We believe that the figure gets "too busy" when every information is added to the graphs.

See above.

Reviewer #3 (Remarks to the Author):

I reiterate my general view that this manuscript demonstrates, quite convincingly, that the nutrient uptake and growth of large chain-forming diatoms is enhanced under turbulent conditions; conditions under which they also form fast sinking aggregates, thus removing both carbon and nutrients from the surface ocean. This is a novel finding that has potentially consequences for our understanding of the oceans' biogeochemistry.

I feel that the authors have adequately addressed my concerns and I recommend the publication of this manuscript.

Responses to reviewer#2.

(The bold, black text from the reviewer includes all comments after the second review and our answers to these are written in red. Our replies to the comments after the first review are written in blue.)

Reviewer #2, Round two, General comments: I thank for the authors for their detailed responses to my review. This study to test the long-standing hypothesis that turbulence stimulates primary production is obviously of strong potential interest considering that it has been sent out for a second round of review despite my initial review. In deference to everyone involved, I have more than doubled my time evaluating this manuscript, both to reconsider my initial evaluation and consider the responses to my initial review. Unfortunately, I cannot give a significantly more positive review, and in fact, the responses to my review comments have provided me with a better understanding of the flaws in the data. The most fundamental problem is that the replication in the experiment was not used for evaluating the central hypothesis, but instead it seems that all data for each treatment were averaged together.

Authors response: The long-standing hypothesis is that turbulence selectively stimulates primary production in large phytoplankton (> 60 μm) – as was stated in l. 105 in the revised version already reviewed by all three reviewers. Turbulence does not stimulate primary production in smaller cells as described in the theoretical background paragraph l. 102-108 in the revised version already reviewed. Unfortunately, reviewer#2 has misunderstood the data analysis. The effect of turbulent shear was first analysed on a community level (bulk rates; GF/F filters analysed by IR-EAMS) with three replicates. We understand that this misunderstanding likely occurred because the data of community PP and nitrate assimilation under turbulent shear (n=3) were indeed not presented in Table 2. We regret this mistake and have added the data to Table 2 and Table 3 in the revised manuscript. However, we did not detect any significant differences in C- and nitrate assimilation at a bulk, mixed community level as a response to turbulence, presumably because the mixed community was dominated by cells which were too small to be sensitive to turbulent shear. The community composition was and is described in the first paragraph of the results. Using SIMS, we were able to reveal that CO₂ assimilation in cells of long diatoms chains, which are instrumental for CO₂ sequestration and large-scale transport through formation of fast-sinking aggregates was indeed sensitive to turbulent shear. We analysed CO₂ assimilation in 572 individual *Skeletonema* cells and in the 306 individual *Chaetoceros* cells and the 102 control cells (incubated without isotopic tracers), and used these measurements in the statistical analysis of the study. We analysed both populations as single cell data (independent of chain size), and as single cells with a fixed position within chains for chains larger than 7 cells. The replicate used to do this was selected as the one which was closest to the average of the bulk nitrate assimilation data. The two chain-forming diatoms investigated accounted for nearly 50% of total nitrate assimilation in the phytoplankton community – as described l. 428-432 in in the revised version already reviewed. We chose the replicate in this way to meet the challenge of measuring statistical differences in in field population with a natural high cell-to cell variance. From each bottle (triplicate) we filtered the bulk community onto individual GF/F filters (triplicate) for bulk community rates measurements as well as filtered cells onto individual GTTP filters (triplicate) for SIMS analysis. This way we always had paired measurements of community assimilation and assimilation by single cells from the very same bottle. This is explained in Methods and we also explained this directly in the previous rebuttal letter.

Furthermore, the statistical methods used are insufficient to account for the multiple comparisons that are made (i.e., single cells versus chains of 2 cells versus chains of 3 cells, up to >7 cells per chain) and to account for the likelihood of non-normal data distributions for subsets of the total data set.

Authors response: we are not sure why the reviewer claims that we did such multiple comparisons (i.e., single cells versus chains of 2 cells versus chains of 3 cells, up to >7 cells per chain) which was certainly not the case. We analysed the whole populations which consisted of ALL chain lengths, which showed no significant response to turbulence for Chaetoceros. Finally, we analysed responses within cell chains as a function of position in cell chains larger than 7 cells chain⁻¹. In these chains we indeed found significant responses to turbulent shear. We believe that our comment that the Chaetoceros population consisted of relatively more shorter (3 to 5 cells chain⁻¹) than long chains (> 7 cells chain⁻¹) (l. 147 in the first revision of the ms) may have led to this misunderstanding. Consequently, we have omitted this sentence in the re-revised ms

The manuscript and characterization of the results are also problematic. The Abstract does not present the nuanced nature of the data, particularly for Chaetoceros, or the primary metrics that the authors are asserting are the basis for support of the central hypothesis (“up to 59%” is not representative).

Author’s response: Unfortunately, we do not agree with the reviewer on this issue. In Chaetoceros, we found zero responses in apical cells and on average 59% increase in central cells within chains. In Skeletonema, we found 32% increase in C-assimilation with turbulent shear. Accordingly, we found up to 59% increase in C-assimilation during turbulent shear. Nature Communications allows for 150 words in the abstract which does not invite for such detailed description of the results. We prefer to give priority to the big picture including small-scale (single cells) and large-scale (sinking aggregates) CO₂ sequestration in the abstract.

The paper does not provide basic information, such as the bulk rate of fixation for the replicates under the different treatments, how C-assimilation differed by chain length, and the contribution of longer chain lengths to overall fixation.

Authors response: We have now included all bulk rate data in Table 2. We have reported all variables in Table 2 under still conditions. Indeed, we should also have included a sentence in the text that we did not detect significant differences in C- or N-assimilation between still conditions and turbulent shear for the whole, mixed community. This important sentence was included in an earlier version of the manuscript, but unfortunately seems to have disappeared during revisions. We have now included data under turbulent shear in both table 2 and table 3, and described the data in the second paragraph of the results (l. 130-132). We have reported the contribution of Skeletonema (2% to 4%) and Chaetoceros (10% to 15%) to total C-assimilation in Table 3. Again, it needs to be emphasized that the total phytoplankton community mostly consist of smaller cells, which do not form aggregates which sink out of the euphotic zone and mixed layer due to their small size. This selectively happens to the large chain-forming diatoms. Their responses in C-assimilation to turbulence are therefore important no matter how much they contribute to the total primary production as also explained in the introduction and the discussion. The more C these cells can assimilate under turbulence in the euphotic zone the more C they can transport to depth when they form aggregates and sink. This is a major insight of the paper.

The absence of this information makes it impossible for the reader put the presented results in context. Basic questions I have include: Is there enough data by chain length to make an evaluation? Is there a trend? What is the justification for making >7 cell chains a subgroup?

Author’s response: As previously explained (in l. 102-108 in the revised version already reviewed), the chains need to have a certain size before they are sensitive to turbulent shear, and to be long enough to form fast-sinking aggregates. We chose to analyse cell chains > 7 cell chains⁻¹ in detail to be able to consider central and apical cells which are well separated by distance. This sensitivity is presumably gradual with increasing chain length but was outside our scope to investigate. In stead, we compared our

measurements of enhancement effects with those predicted by mass transfer theory and indeed found very good agreement between theory and measurements.

What are the subgroups' contributions to total C assimilation? Are longer chains more productive?

Authors response: As mentioned above, we did not divide our data in to subgroups beyond chains > 7 cells. The interesting point is that C-assimilation in the longer cell chains which can coagulate and sink out of the euphotic zone is sensitive to turbulence. All chains lengths contributed 12% to 20% (Table 3) to the total bulk C-assimilation which is mainly composed of smaller, non-sinking cells. However, the cell chains represent the fraction of the phytoplankton community which eventually forms fast-sinking aggregates and sediment out of the euphotic zone, and thus transport fixed CO₂ to depth of the ocean. This was also pointed out in the final, concluding paragraph l. 355-366 in the revised manuscript already reviewed.

(I also wonder if the data suggest other related or unrelated hypotheses that others might want to pursue, such as, do the data provide information related to cell division?) I also find it unnecessarily difficult to determine in which cases there are statistically significant effects in Figures 4, 5 & 6.

Authors response: Yes, the C-specific C assimilation and N-specific N assimilation are related to cell growth as was explained l. 162-164 in the revised manuscript already reviewed (we also have unpubl. measurements of SIMS and cell division rates performed in cultures). Statistically different values cannot be judge by eye when samples are large. We have written through-out the manuscript where values were significantly different or not to each other.

Overall, I am concerned that the data as they stand present weak and poorly characterized support for the central hypothesis. From my perspective, when one is claiming to demonstrate a paradigm, the experimental design and execution and the data presentation and evaluation need to be rigorous, and the results need to be unambiguous. Unfortunately, for the reasons laid out above, this work does not meet this standard. However, I would consider this study publishable even if the results are ambiguous, if the data were more rigorously handled and the results were described in the Abstract in a balanced way.

Author's response: Unfortunately we do not agree on this statement and judgement of our study and manuscript for the reasons we have given above as answers to each concern raised by the reviewer. Not only have we, for the first time, directly measured CO₂ and nitrate assimilation in field populations of diatom cell chains but we have also compared our results with those expected from mass transfer theory and demonstrated their formation of aggregates.

For lack of a better place to note this, in Lines 141-143 I think there is a mistake here, because in Figure 5, it does not look to me that Skeletonema had 32% higher C assimilation under turbulent conditions. Is this supposed to be "C:N assimilation"? That would make more sense since this statement is associated with the C:N discussion above. Whatever the answer, this would good information for the Abstract.

Author's response: The Skeletonema data are the open bars. Under stagnant conditions the cell-specific C-assimilation rate is 18.5 fmol C cell⁻¹ h⁻¹ and under turbulent shear it is 24.7 fmol C cell⁻¹ h⁻¹. This is equal to 32% increase during turbulent shear.

My responses are below.

Reviewer #2 (Remarks to the Author): This study seeks to provide single cell data to test the hypothesis that that primary production in chain forming diatoms is stimulated by turbulence. I was asked to review the application of SIMS in this study, and therefore most of my comments are from that perspective.

I found this a poorly organized and incompletely presented manuscript, and therefore it was very difficult to evaluate, even within my limited scope. The SIMS results are not well organized and are hard to reconcile with each other, particularly on the point of C-assimilation by Chaetoceros. The figures lack basic reference information, and Fig. 4 in particular is hard to interpret and reconcile with the narrative. The Method Section does not include important information and presents the work as centered on two experiments, which is not how the research is presented in the Results. Finally, the conclusion that turbulence stimulates small-scale biological CO₂ assimilation (L40-41) does not seem to follow from the results: the narrative associated with Fig. 4 states that there is not a turbulence treatment effect for C-assimilation by Chaetoceros (L149-151), though the cell chain results make that result hard to understand (L185-187). I did not analyze the N-assimilation results in detail given the substantive problems with the manuscript.

Response: Fig. 4 shows values for the whole populations of diatoms, i.e., all cells in short or long chains. We have made this clearer in the revised manuscript (l. 146-147). A closer analysis showed that C-assimilation is increased by turbulent shear in the cells of the long chains (>7 cells per chain) (Fig. 6) and these long chains also form sinking aggregates due to their large size.

Reviewer Response: The statistical justification for separating out chains of >7 cells is not presented, the data are not handled based on the replicates, the statistics used are not justified for these small sample numbers, and the quantitative importance of the observed effect is not presented.

Author response: As explained earlier in this response letter as well as in the theoretical paragraph (L.84-108 and Eq. 3) sensitivity to turbulence depends on cell size or chain size. We did not observe significant sensitivity to turbulence in the Chaetoceros population encompassing ALL chain lengths (Fig. 4, l. 146-1.147) also not in the total plankton community measured by IRMS (now stated l. 130-132 in the re-revised manuscript) because it is composed by far more small cells than large cells. However, in Chaetoceros chains with 7 cells or larger we did measure significant responses to turbulence (Fig. 6), and it is the long chains that form aggregates and sink to depth.

Methodologically, I was concerned that the statistical methods were not discussed in the Method section, particularly because it was not clear how the replicates were handled. I was also concerned about standardization of the measurements, use of the data for the control samples, and measurement precision.

Response: It is important to remember that we are dealing with natural mixed field-populations of diatoms, and this is one of the innovations in this study. Thus, it is not straightforward to compare bulk measurements of the mixed phytoplankton community with single-cell measurements within one genera. Our analysis showed that the chain-forming diatoms contributed 46% of total nitrate assimilation in the community but only 12% of total C-assimilation because all phytoplankton assimilate C but not all phytoplankton can assimilate nitrate (Table 3). The cell-to-cell variability in nitrate assimilation rates was much larger (the standard deviation was ca. 50% of the mean value; Fig.5, and Fig. 6) than the variation of the average of nitrate-assimilation at the community level (std was ca. 10% of the mean value; Table 2). We measured a large number of cells to obtain good statistics at the single cell level in the replicates which were closest to the average value of bulk nitrate assimilation rates by the community (GF/F filters analysed by IR-EAMS)(l. 431-433). We analyzed increasing number of cells until the average assimilation rates calculated were stable and representative for the population (l. 242-243 and l. 460-463). In the revised manuscript, we have shown examples of such analysis for Skeletonema and Chaetoceros (Fig.8) which clearly shows that average values vary largely when the number of cells analysed are below ca. 50, but it stabilizes and reaches representative average values when the number of cells analysed is > 50. We feel confident that our data are indeed representative average values even though the cell-to-cell variation is large. Subsequently, we have used student-T tests, which is valid for comparisons of two populations with a normal distribution and a large number of observations (n). We have reported all relevant numbers of the statistics in brackets throughout the text. The control samples were used to calculate excess isotopic ratios of ¹³C:¹²C and ¹⁵N:¹⁴N. We have also specified this in the text (l. 401-403 and l. 467- 468). The

high cell-to-cell variation in field populations is important to document in a biological and evolutionary context and is now included in the discussion (l.236-251).

Reviewer Response: The Method section still lacks details on the use of the replicates, the statistical tests and the use of the control samples and standardization. Based on the statement above, the authors combined the data from the replicates, which reduces the experiment to an n of one, which is not a good basis for claiming to demonstrate a paradigm. The authors should re-analyze the data, separating out the replicates, and which should then be statistically analyzed, taking into account whether the data are normally distributed and the fact that there are multiple comparisons. **Author**

response: As stated above and in the manuscript: *the variation of the average of nitrate-assimilation at the bulk community level measured by IRMS was less than the cell-to cell variation measured in Skeletonema and Chaetoceros. We did SIMS analysis the replicate closest to the mean value measured by EA-IRMS.*

*Using SIMS we measured a large number of cells ($50 < n < 200$) to encompass the cell-to-cell variation and to reach a representative mean value (Fig. 8) We did not pool the data from the different replicates as the reviewer mistakenly presume. In fact, there are also no multiple comparisons as the reviewer mistakenly believes. In the previous versions of the manuscript we used student's t-test to compare various cells. As recommended by the editor, we have consulted an expert in statistics and done even more statistical analysis. We have included a paragraph on the statistics in the re-revised manuscript l. 532-547: “the significance of differences in the carbon and nitrogen assimilation data at the community level (EA-IRMS data) were analysed using two way ANOVA test in SigmaPlot version 11. The replicates of cell-specific C and N assimilation rates measured by SIMS were selected from the bottles with community nitrate assimilation rates closest to the mean value the nitrate assimilation measured by EAIRMS under still conditions and turbulent shear, respectively. Increasing numbers of the cellular C- and N-assimilation rates measured by SIMS were calculated until the average values per cell were stable and the standard error was <5% of the average value to achieve representative average values for single cells of both genera of diatoms (Fig. 8). The C and N assimilation rates calculated from the $^{13}\text{C}/^{12}\text{C}$ and $^{15}\text{N}/^{14}\text{N}$ ratios of the 572 individual *Skeletonema* cells and of the 306 individual *Chaetoceros* cells and the 102 control cells were used in the statistical analysis. The C assimilation rates followed a normal distribution as tested in detail using statistical analysis software (SAS) of their residuals. The C and N assimilation data in the sub-groups of cells were also described by normal distributions and (non)-significant differences between these were analysed by two way ANOVA test in SigmaPlot version 11 as well as by student's t-test in Microsoft Excell (2010). These two different statistical analyses gave similar results. “Below, please find the distribution of C-specific C-assimilation as well as the distribution of the residuals are shown (analysis performed by Senior lecturer and consultant Kerstin Wiklander, Applied Mathematics and Statistics at Department of Mathematical Sciences, University of Gothenburg):*

Author response: As stated above and in the manuscript: *the variation of the average of nitrate-assimilation at the bulk community level measured by IRMS was less than the cell-to cell variation measured in Skeletonema and Chaetoceros. We did SIMS analysis the replicate closest to the mean value measured by EA-IRMS. Using SIMS we measured a large number of cells ($50 < n < 200$) to encompass the cell-to-cell variation and to reach a representative mean value (Fig. 8) We did not pool the data from the different replicates as the reviewer mistakenly presume. In fact, there are also no multiple comparisons as the reviewer mistakenly believes. In the previous versions of the manuscript we used student's t-test to compare various cells. As recommended by the editor, we have consulted an expert in statistics and done even more statistical analysis. We have included a paragraph on the statistics in the re-revised manuscript l. 532-547: “the significance of differences in the carbon and nitrogen assimilation data at the community level (EA-IRMS data) were analysed using two way ANOVA test in SigmaPlot version 11. The replicates of cell-specific C and N assimilation rates measured by SIMS were selected from the bottles with community nitrate assimilation rates closest to the mean value the nitrate assimilation measured by EAIRMS under still conditions and turbulent shear, respectively. Increasing numbers of the cellular C- and N-assimilation rates measured by SIMS were calculated until the average values per cell were stable and the standard error was <5% of the average value to achieve representative average values for single cells of both genera of diatoms (Fig. 8). The C and N assimilation rates calculated from the $^{13}\text{C}/^{12}\text{C}$ and $^{15}\text{N}/^{14}\text{N}$ ratios of the 572 individual *Skeletonema* cells and of the 306 individual *Chaetoceros* cells and the 102 control cells were used in the statistical analysis. The C assimilation rates followed a normal distribution as tested in detail using statistical analysis software (SAS) of their residuals. The C and N assimilation data in the sub-groups of cells were also described by normal distributions and (non)-significant differences between these were analysed by two way ANOVA test in SigmaPlot version 11 as well as by student's t-test in Microsoft Excell (2010). These two different statistical analyses gave similar results. “Below, please find the distribution of C-specific C-assimilation as well as the distribution of the residuals are shown (analysis performed by Senior lecturer and consultant Kerstin Wiklander, Applied Mathematics and Statistics at Department of Mathematical Sciences, University of Gothenburg):*

In the revised manuscript, we have now reported statistics based on two way ANOVA as advised by the expert (Senior lecturer and consultant Kerstin Wiklander, Applied Mathematics and Statistics at Department of Mathematical Sciences, University of Gothenburg). Below, we here show the analysis of C-assimilation in Skeletonema and Chaetoceros under still conditions and turbulent shear (the key data which show significant responses to turbulence in the study). Resultat från SAS performed by Kerstin Wiklander:

The SAS System

The Mixed Procedure

Class Level Information

Class	Levels	Values
Vatten	2	still turb
Art	2	c s

Covariance Parameter Estimates

Cov Parm	Estimate
Residual	0.000028

Fit Statistics

-2 Res Log Likelihood	-4550.0
AIC (smaller is better)	-4548.0
AICC (smaller is better)	-4548.0
BIC (smaller is better)	-4543.6

Type 3 Tests of Fixed Effects

Effect	Num DF	Den DF	F Value	Pr > F
Vatten	1	598	11.56	0.0007
Art	1	598	22.66	<.0001
Vatten*Art	1	598	1.93	0.1656

Least Squares Means

Effect	Vatten	Art	Estimate	Standard Error	DF	t Value	Pr > t
Vatten	Still		0.009053	0.000364	598	24.86	<.0001
Vatten	Turb		0.01070	0.000321	598	33.32	<.0001
Art		C	0.01103	0.000407	598	27.12	<.0001
Art		S	0.008723	0.000265	598	32.91	<.0001
Vatten*Art	Still	C	0.01055	0.000648	598	16.28	<.0001
Vatten*Art	Still	S	0.007560	0.000333	598	22.72	<.0001
Vatten*Art	Turb	C	0.01152	0.000492	598	23.41	<.0001
Vatten*Art	Turb	S	0.009885	0.000413	598	23.95	<.0001

Differences of Least Squares Means

Effect	Vatten	Art	Vatten	Art	Estimate	Standard Error	DF	t Value	Pr > t
Vatten	Still		Turb		-0.00165	0.000486	598	-3.40	0.0007
Art		C		s	0.002311	0.000486	598	4.76	<.0001

Differences of Least Squares Means

Effect	Vatten	Art	Vatten	Art	Estimate	Standard Error	DF	t Value	Pr > t
Vatten*Art	Still	C	Still	s	0.002985	0.000728	598	4.10	<.0001
Vatten*Art	Still	C	Turb	c	-0.00098	0.000814	598	-1.20	0.2302
Vatten*Art	Still	C	Turb	s	0.000660	0.000768	598	0.86	0.3904
Vatten*Art	Still	S	Turb	c	-0.00396	0.000594	598	-6.67	<.0001
Vatten*Art	Still	S	Turb	s	-0.00233	0.000530	598	-4.39	<.0001
Vatten*Art	Turb	C	Turb	s	0.001637	0.000642	598	2.55	0.0111

Chaetoceros Carbon End-cells vs mid cells vs turbulence vs still

Two Way Analysis of Variance

Monday, February 19, 2018, 1:48:24 PM

Data source: Data 1 in Notebook1

General Linear Model

Dependent Variable: Col 3

Normality Test: Passed (P = 0.408)

Equal Variance Test: Passed (P = 0.297)

Source of Variation	DF	SS	MS	F	P
Col 1	1	0.000164	0.000164	9.642	0.003
Col 2	1	0.0000593	0.0000593	3.484	0.067
Col 1 x Col 2	1	0.0000415	0.0000415	2.440	0.124
Residual	54	0.000919	0.0000170		
Total	57	0.00118	0.0000206		

The difference in the mean values among the different levels of Col 1 is greater than would be expected by chance after allowing for effects of differences in Col 2. There is a statistically significant difference (P = 0.003). To isolate which group(s) differ from the others use a multiple comparison procedure.

The difference in the mean values among the different levels of Col 2 is not great enough to exclude the possibility that the difference is just due to random sampling variability after allowing for the effects of differences in Col 1. There is not a statistically significant difference (P = 0.067).

The effect of different levels of Col 1 does not depend on what level of Col 2 is present. There is not a statistically significant interaction between Col 1 and Col 2. (P = 0.124)

Power of performed test with alpha = 0.0500: for Col 1 : 0.839

Power of performed test with alpha = 0.0500: for Col 2 : 0.324

Power of performed test with alpha = 0.0500: for Col 1 x Col 2 : 0.204

Least square means for Col 1 :

Group	Mean	SEM
C-end	0.0115	0.000780
C-mid	0.00816	0.000755

Least square means for Col 2 :

Group	Mean	SEM
still	0.00883	0.000780
turb	0.0109	0.000755

Least square means for Col 1 x Col 2 :

Group	Mean	SEM
C-end x still	0.0114	0.00110
C-end x turb	0.0117	0.00110
C-mid x still	0.00630	0.00110
C-mid x turb	0.0100	0.00103

All Pairwise Multiple Comparison Procedures (Holm-Sidak method):

Overall significance level = 0.05

Comparisons for factor: **Col 1**

Comparison	Diff of Means	t	Unadjusted P	Critical Level	Significant?
C-end vs. C-mid	0.00337	3.105	0.003	0.050	Yes

Finally, based on the SIMS Methods section, the mass resolving power used was not sufficient to resolve what is typically a significant isobaric interference at 27 amu (11B 16O- on 13C 14N-), which would affect the 13C/12C measurement. This interference is likely the cause of the background in the 13C/12C images (see below), and therefore the TEP interpretation should be reconsidered.

Response: We agree with the reviewer that part of this signal may be due to a small interference from BO from the pre-spattered frustules and we have now omitted to mention TEP in this context.

Reviewer: Okay.

BO- would also be an interference in the measurements on the cells, but it is potentially not significant when the CN- count rate is high. Standard and control measurements are necessary to demonstrate that the cellular 13C/12C data are valid. These issues will need to be considered in the statistical analysis. If there were other 13C/12C measurements (e.g., using C dimers or monomers), that could be useful.

Response: We are aware that traces of BO are located in the frustule of diatom cells and may interfere with the 13C/12C when measurements are not performed in the cell interior, but in or close to the frustule. We have now included the reference by Mejía et al (2013) concerning this issue. The isotopic images (Fig. 2) and the fact that C:N assimilation ratios were close to Redfield demonstrate that our measurements were indeed within the cells where CN-counts are high. Controls (with no added isotopic tracers of 13C-bicarbonate) indeed showed slightly higher 13C:12C ratios of 0.0112-0.0116 (relative to the natural abundance of 0.0109) which may be due to interference from BO. These control measurements of 13C/12C in un-amended samples (including a potential signal from BO) were subtracted from the value measured under the 13C-bicarbonate-amended condition to calculate excess 13C/12C which should be true values of excess 13C/12C (l. 465-468).

Reviewer Response: While I agree that the data appear valid, I think it is necessary that the flaws in the instrument set up be made explicit and up front in the Methods section. I do not want this paper to become a reference for how to make these measurements, especially if published in a Nature journal. It has long been established that boron is a ubiquitous surface contaminant that produces BO- and can interfere with CN- measurements at mass 27 (Zinner et al., 1989, Geochimica et Cosmochimica Acta 53, 3273–3290). Furthermore, 6000 MRP is only marginally high enough to resolve 13C2- from 12C14N-, which is unimportant in natural samples, but which can become significant in 13C-labeled samples or low nitrogen samples (Zinner et al., 1989 is a reasonable

reference. Unfortunately, Peteranderl and Lechene (J Am Soc Mass Spectrom 2004, 15, 478–485) mislabel the $^{13}\text{C}_2$ peak, though they correctly present the required MRP to be ~ 7000). Therefore, the text in the Methods section should clearly acknowledge that the mass resolving power used was in error—concurrent with the explanation for why these species were used. Starting from the text on line 465, I propose something like: <<For each cell chain we recorded secondary ion (SIMS) images of $^{13}\text{C}^{14}\text{N}^-$ and $^{12}\text{C}^{14}\text{N}^-$, and $^{12}\text{C}^{15}\text{N}^-$ using a peak-switching routine at a mass resolution of ca. 6000 (M/ Δ M), which is less than the $\sim 12,000$ M/ Δ M necessary to resolve BO- from $^{13}\text{C}^{14}\text{N}^-$ (Zinner et al., 1989); BO- is a common surface contaminant and present in the diatom frustules [ref]. Also, ~ 7000 M/ Δ M is necessary to fully resolve $^{12}\text{C}^{14}\text{N}^-$ from $^{13}\text{C}^{13}\text{C}^-$, which can be significant for highly ^{13}C -enriched samples. Measurements on control samples suggest that these interferences did not significantly affect the isotope measurements on the cells, but BO- did affect ratios from low yield regions. The use of CN isotopomers have the advantage that they are bright and two isotope ratios can be measured from measurements of three isotopomers.>>

Response: Thank you very much. We included this text l. 448-457.

Given the analytical issues, the standard and control data should at least be presented in a supplemental section, and the uncorrected statistics (mean, standard deviation and n) should at least be presented in the Methods section, and the range of measured ^{13}C enrichment should be reported.

Authors response: we have included these l. 470-472.

As a general point, based on the method section, there were bulk IRMS data for these experiments, but I could not find them in the manuscript. Given the variability in the SIMS data, bulk C and N assimilation data would be a useful reference for the single cell data.

Response: Please see earlier response. The community (bulk) C and N assimilation data were presented in Table 2 in the original as well as in the revised manuscript. As mentioned earlier, the cell-to-cell variability in nitrate-assimilation rate is much larger (std was ca. 50% of the mean value) than the variation of the average of nitrate-assimilation on a community level (std was ca. 10% of the mean value; Table 2). Thus, variability of IRMS data was low whereas cell-to-cell variability was high. However, our statistical analysis shows that we have analysed enough cells to achieve representative mean values (Fig. 8). We may also mention here that in a N_2 -fixation study during the cyanobacterial summer bloom in the Baltic Sea, we found a 1:1 ratio between IRMS data and SIMS data because N_2 -fixation was confined to a well-defined group of cyanobacteria (Klawonn et al., 2016).

Reviewer Response: This is helpful because I did not understand how the IRMS data were used, but I am still confused by why the results for “still” and “turbulent” treatment data aren’t compared. Based on the author response, the treatment effect should be detected in the bulk data, and if this is true, it would be a substantial piece of evidence showing the significance of the hypothesized effect. However, if the effect is not measurable in the bulk data, it should still be presented in Table 2. Also, uncertainties should be calculated for C:N and C:other ratios using Gaussian error propagation and presented in Table 2.

Authors response: No, the reviewer has unfortunately misunderstood this matter. As mentioned earlier, we have explained that no significant responses to turbulence can be detected in the bulk data (=whole plankton community) presumably because they represent much more small (non-sinking) cells compared to large cells. The bulk community includes many other phytoplankton than the large chain-forming diatoms (l. 114-120 in the revised manuscript already reviewed). The chain-forming diatoms contributed 12% to 20% to total C-assimilation was described in Table 3 in the revised manuscript already reviewed as well as in the original submission. As mentioned earlier, we show that the C- assimilation in large diatoms which form aggregates and eventually sink out to depth is indeed increased during turbulence in the euphotic zone. The more C these cells assimilate the more they can transport to depth – CO_2 sequestration to depth is not a function of bulk C-assimilation in the whole community. We do not understand how Gaussian error propagation would improve the manuscript and its message.

My detailed comments follow:

L66: The description of SIMS does not allow for differences among SIMS instruments. It would be better to be more specific about the critical properties for this study (e.g., imaging, dynamic, magnetic sector).

Response: NanoSIMS and IMS1280 are both dynamic SIMS, with similar magnetic sector mass separation and imaging capabilities but different in spatial resolution (50–100 nm for nanoSIMS compared to ca. 1 μm for IMS1280). The lower spatial resolution of the IMS1280 allows for higher through-put compared to nanoSIMS when cells are large (> 3 micrometer). We have added this information in M&M section (l. 437-439).

Response of reviewer: I would suggest taking a step back on the instrument explanation, given that TOF-SIMS instruments are more common than dynamic SIMS instruments and this manuscript is aimed at a broad audience. A key point of the instrument used is that it has high mass resolving power with flat-top peaks, which enables high precision and accuracy isotope measurements. The argument that these analyses are faster than nanoSIMS, if true, would be interesting, but the authors have not substantiated this claim, and the method description is missing key information. Lower spatial resolution, per se, does not allow for higher throughput, and higher beam current (which the authors seem to be associating with lower spatial) does not necessarily allow for higher throughput either. The analysis currents used in this study can easily be achieved with a NanoSIMS 50L. For these analyses to be substantially faster, the difference would have to be the detector, which is not specified in the manuscript. Typically, imaging on any dynamic SIMS instrument is performed with electron multipliers because Faraday cups are too slow. In turn, it is the electron multipliers that tend to limit the speed of the analysis because analyst limit the secondary ion count rate because high count rates result in premature detector failure. It would be very interesting if the detector used was an EM that allows for high (>1 000 000 counts per second) secondary ion currents or if the analyses was somehow performed with a Faraday cup (though this would not likely be faster because of the slow Faraday cup response). However, if the analyses were performed with standard electron multipliers, the analysis time would be similar for the IMS 1280 and a NanoSIMS. In any case, the authors should specify the detector used and the secondary ion currents obtained since that is the critical part of the analysis speed.

On the topic of speed, there is a problem with the dwell time given in the Methods (L454: “The dwell times were 1 s, 5 s and 2 s...”). My guess is that the units should be milliseconds, not seconds, as seconds would yield very long analyses. Also, please write “dwell time per pixel” and “ms/pixel” for clarity sake.

Authors response: We do not understand why we should include explanation of TOF-SIMS which is not an appropriate nor an alternative instrument to use for these type of investigations, and we do not think the broad audience has interest in such comparisons. We argue that SIMS has a higher throughput simply because of its lower but sufficient spatial resolution. nanoSIMS with a spatial resolution of 50 nm is unnecessary when analyzing large cells. It takes longer time to map a cell with 50 nm resolution compared to 1 μm resolution. In other respects, nanosims and SIMS are comparable techniques.

L134: The use of the term “stable isotope” in context of the images implies that these images are different from other secondary ion images, which they are not. I suggest referring to the image data as ion images and ion ratio images or possibly SIMS images.

Response: We have corrected this throughout the manuscript as suggested by the reviewer.

Okay

L135: I suggest giving a basic statement on how the image data are used: e.g., isotope ratio data are extracted from individual cells based on cell morphology visualized in the CN ion images.

Response: We have clarified this in the revised manuscript. The cell chains are easy to recognize and distinguish in the $^{12}\text{C}^{14}\text{N}$ ion images (Fig. 2: and l. 455-456). As mentioned below, cell sizes were measured by light microscopy in parallel samples (l. 387-391).

Okay

L135-138: The explanation of the difference in relative ^{15}N and ^{13}C enrichment in the TEP is not satisfactory. These ratios reflect the source of the N and C (here, old versus new), not the concentration of each element in the material. Taken on its face, higher ^{13}C enrichment relative to ^{15}N enrichment suggests that the C is new and the N is not. However, based on the reported mass resolving power for the analyses and the species monitored, it is likely that the area around cells in $^{13}\text{C}/^{12}\text{C}$ images reflects a background count rate of $^{11}\text{B}^{16}\text{O}^-$ at 27 amu (see below). This is even more likely since the CN- count rate in the region around the cells is very low (i.e., BO- counts are more likely to be significant). It would be useful to have data for a reference material, such as the substrate around unlabeled diatoms, to rule out this possibility.

Response: It is well-known that Chaetoceros and Skeletonema produce TEP under nutrient limitation (Kiørboe and Hansen, 1983) as also stated in the original submission. However, we agree that Boron, which is a trace nutrient for diatoms, and which is mostly located in the silicified cell walls may be partly responsible for the apparently high counts on the filter due to sputtered material from the silicate frustules. We have rephrased the text accordingly.

Okay

If this is an interference issue, it would not necessarily make the data set for the cells (high CN regions) useless, but the uncertainty on the measurements would have to be re-evaluated.

Response: As mentioned earlier, Boron is a micronutrient and it is mostly located in the silicified cell wall which is pre-sputtered away before the actual measurements are done in the cell interior (with high CN) (Meija et al., 2013). We subtracted the slightly higher $^{13}\text{C}/^{12}\text{C}$ value measured in our control samples when calculating rates. Hence, our data are corrected for this small potential interference by Boron. Cell-specific rates vary largely both in field populations and between different strains of the same species of phytoplankton. SIMS data measured with higher mass resolution (MRP = 11000) which should resolve BO and CN counts on Skeletonema cultures have shown similar cell-specific C-assimilation rates and C:N assimilation ratio as reported in this manuscript when the control measurements were similar to those of the natural abundance of $^{13}\text{C}:^{12}\text{C} = 0.0109$ (own unpubl. data). Hence, we feel confident of our data quality and interpretation.

Reviewer response: If I understand this correctly, the SIMS data for the one control sample were used to calculate the single cell assimilation rates using equation 1. The data for these should be reported in the paper, presumably in the SIMS method section.

Author response: Yes, the cell assimilation rates were calculated from excess enrichments in cells with added tracers relative to the controls (natural abundances when no tracers were added). Controls do not vary much as compared to amended samples and are now reported with its mean value and std. The average $^{13}\text{C}/^{12}\text{C}$ in un-amended samples (controls) and the std was 0.0113 ± 0.0006 (n=102), whereas that of amended samples was 0.0143 ± 0.0017 (n=878). This information has been added to M&M I. 472-474.

If the result stands, I would suggest that this discussion not come as the first note on the images since this is an interpretation. I also want to note that I do not think that it is obvious that the images can be directly interpreted for relative enrichment. To the extent that the authors discuss their interpretation of the TEP enrichment, I think it would be better to make the interpretation based on numerical data.

Response: As noticed earlier, we have omitted this comment on the images in the results as suggested by the reviewer. TEP production was never focus of our study.

Okay

At the risk of causing confusion, I'll just note here that the way the authors used the word "background" in this context is confusing. The word "background" implies counts that are not from the sample, whereas the authors seem to mean counts that are not directly from the diatoms themselves, but rather from the TEP.

Response: The reviewer is right. We have omitted the word “background” when commenting *on the images as suggested by the reviewer*.

Okay

This, however, is beside the point if BO- is the issue.

Response: please see comment on BO above.

Okay

L132: Please be more explicit about the presentation of the nitrate and ammonium experiment data. Based on the Methods section, there are two sets of experiments, but this section does not organize the results in that way, making it more difficult than necessary to relate them to each other.

Response: The nitrate and ammonium experiment data on non-aggregated diatoms are presented in Figure 5. The N-data are discussed relative to the C-assimilation data which were not significantly different in the two experiments (in situ versus laboratory) presumably because light intensity was high enough to saturate photosynthesis and no photoinhibition occurred as intended (l. 151-154).

Reviewer Response: My comment here was at a more general level. I found the nitrogen assimilation data poorly integrated in general. I would even refer it back to the Abstract, in which the paper is framed in terms of primary production, and nitrogen assimilation seems tacked on.

Author response: Unfortunately, we do not agree with the reviewer on this matter. In fact, C:N fixation ratios were close to Redfield ratio. Further, export production (e.g., sedimentation of aggregated diatoms) is supposed to be limited by nitrate while regenerated production by smaller phytoplankton is limited by ammonium. C incorporated into cells is linked via nutrient supply so these are tightly coupled as also was explained L. 301-309 in the revised manuscript already reviewed. We even compared measured nitrate assimilation with diffusion-limited nitrate fluxes according to mass transfer theory and found those to be similar.

L143: Are the numbers mean +/- 2 standard errors of the mean?

Response: it is +/- 1 standard error of the mean (l.139).

Okay

L147: How do the replicates compare? How are they handled in the statistical test?

Response: Please see earlier explanation.

Reviewer Response: As noted above, it is normal and expected in ecology for there to be replication to control for natural variability, which is often high. In the Method section, the authors state that they performed replication (L398-400), but I cannot find any presentation of how the replication is used. My impression from the paper and the Author Response is that they combine the data from all of the replicates together, which would make the results essentially anecdotal. To provide a more robust test of the central hypothesis, the treatments should have replication.

Authors response: we calculated mean values and std of the mean values of the bulk rates and reported these in Table 2. As explained earlier in both this letter and the earlier letter to the reviewer, the replicate which was closest to the average value of nitrate assimilation in the bulk was further analysed by SIMS. We analysed a large number of cells in the replicate to meet the larger variation of assimilation rates at a single cell level. We did not combine data from all of the replicates together.

L167: Is “C-specific C-assimilation” common term?

Response: this term has been used in several NanoSIMS and SIMS studies by now and has the advantage that it is a proxy for growth independent of cell size as we have explained in the text (l. 162-164).

Okay

L180-191: It is surprising that there is a significant treatment effect for the Chaetoceros cells in the interior of the chains, but not effect overall. Does this mean that most of the cells discussed in the previous section were primarily not in chains? At the start of that section, on L134, it is stated that the data were “measured in single cells within individual cell chains.” I am guessing this is just a clarity of writing issue.

Response: Apical cells have a larger surface area exposed to ambient water. The different responses of Skeletonema and Chaetoceros may be explained by different morphology and flexibility of the chains (l. 286-289). Diffusion and shear are size-dependent processes (Eq. 2 and Eq.3) as described in the theoretical paragraph (l.84-108). A significant effect of turbulent shear was measured in cell chains with more than 7 cells (l. 179) and in the figure legend Fig. 6). In fact, this finding is the first of its kind on natural diatom field populations. The data points in Figure 4 represent all measurements performed at a single cell level in chains of various length (≥ 3 cells chain -1). The entire population consists of cells in cell chains. We did not observe a significant effect of shear in the whole diatom population but only on the longer chains which also form the sinking aggregates (l. 336-344).

Reviewer Response: This explanation of the results makes sense to me. If this result holds up under a more rigorous treatment of the data, I would suggest putting this level of subtly into the Abstract to set up the paper. As the Abstract stands, these results seem inconsistent with the broad brush claim of demonstrating that turbulent shear increases primary productivity.

Authors response: unfortunately, we do not agree with the reviewer that such subtle detail should be included in the abstract with a word limit of 150 words. We have never claimed that turbulence increase PP in general. We have demonstrated that turbulence increases C-assimilation in chain-forming diatoms.

L241: I don't understand. This statement seems to contradict L149-151, which state that there was no statistical difference in cell-specific C-assimilation rates for Chaetoceros.

Response: Please see answer above.

Okay.

L280-282: Unless the authors can produce a high resolution mass spectrum demonstrating that 11B 16O- was resolved from 13C 14N- at mass 27 amu, this line cannot be supported. Also, it would be more convincing if there were significant counts of the species used for the presumed TEP measurements.

Response: We agree that pre-sputtered BO-isotopes (trace nutrient) from the silica frustule may have interfered here and we have omitted the TEP elaboration from the manuscript.

Okay

L357: The meaning of "final labeling" is not clear. Does this mean 5% of the bicarbonate in the incubation was labeled? If that is the case, it is relevant to know if this is the calculated abundance of 13C, given that 13C is naturally a 1% isotope.

Response: We have clarified this and explained that the excess 13C:12C ratio of bicarbonate was 0.036 (l. 395). The uptake rates are calculated on basis of the excess 13C:12C ratio of bicarbonate in the ambient water, the excess 13C:12C of the OM, and the incubation time (Eq. 1).

Okay

L361-362: Please be more precise. Presumably the authors mean: "and one 1L bottle was incubated with natural, unlabeled NH₄⁺ at the same concentration as the other bottles..."

Response: The control bottles were not amended (neither with 15N- nor with 14N-ammonium) to measure the natural isotope ratios of 13C/12C and 15N/14N in the diatoms cells as well as in the phytoplankton community. We have clarified this in the text (l. 401-403).

Okay

L363-364: If there were nine bottles and two sampling points at which times 3 bottles were sampled, what happened to the last three bottles? Was there a 3rd sampling point? (see question below: was it 2 and 5 hours?) When was the control sampled?

Response: We had three sampling points: T0, T2 and T5. The control was sampled at T5 (but the isotope ratios should be constant at their natural abundance during incubation as this was not amended with isotopic tracers) (l. 404).

Reviewer Response: I am sorry if I missed this explanation in the initial manuscript, but as you might imagine, this leaves me with the question, How are these time points used in the manuscript? I do not see any indication of time in any of the graphs. Showing a temporal trend would support the central hypothesis. Please include some explanation in the manuscript or remove mention of the time points if not used.

Author's response: We measured different time points during all incubations in order to calculate RATES which are reported in Fig. 4, Fig. 5, Fig. 6 and Fig. 8, and in Table 2.

L364: 25 h or 2.5 h or something else?

Response: 2 and 5h (l. 404).

Okay.

L373: The phrasing here raises the question of whether the T0 samples had been exposed to labels, but immediately sampled, or were not exposed to label. Based on the description above, I understood that it was the latter. If that is the case, "At the start" would be a better way of stating.

Response: yes, it was the latter: "Triplicate incubation bottles without tracer were stopped by filtration at T₀ and after 2 and 5 h" (l. 404).

Okay.

L385: In the SIMS method section, no mention is made of standardization of the analyses. Please include.

Response: We have run controls on cells which were not exposed to isotopic tracers to compare these with natural abundances. Please see comment above. Considering the standardization procedure of pre-sputtering and measurements please see the following comment below

Reviewer Response: If I am not missing it, the T0 data should be reported separately from the unlabeled control.

Author's response: T0 data can in practice be difficult to achieve because filtration takes time when T0 and T1 are close. However, T0 were not significantly different to the controls in our study.

L391: There is insufficient information to assess the analysis method. What is the basis for stating that silica is removed but the cellular material remains? Do they have an estimate of the depth of sputtering?

Response: The image made without sputtering shows a CN depletion until the Si has been removed. After this the cellular material with high CN is clearly visible as can be seen on the isotope image (Fig. 2) (l. 445-455).

Reviewer Response: This would be good information to include in the Method section.

Author's response: It was included in the Method section L. 445-455 in the revised manuscript already reviewed.

What was the area sputtered with the 10 nA beam?

Response: Slightly larger than the imaged area in order to eliminate possible slight offsets between the two beams (sputter and analytical) and "edge effects" (l. 446-448).

Reviewer Response: The information above is very helpful, but still it is better to be more precise, if only giving a range.

Why "< 5min"?

Response: We calibrated the time needed based on appearance of the cellular material, with a bit of overhead to allow for variations in natural samples – after that we sputtered at a fixed time of 5 min. We have omitted the “<” in the revised manuscript (l. 445).

Okay.

Why not a more specific sputtering period?

Response: Our test runs showed that we needed 5 min to get through the silica frustule and the interior cell material with high CN content could be imaged. We have clarified this in the revised manuscript (l. 448-450).

Okay

Was there a specific marker used to determine that the correct depth was reached?

Response: No, analyses were automated into cells after a fixed period needed to reach the interior of cells with high CN had been assigned (l. 448-450).

Okay

Where the cells imaged by SEM after sputtering?

Response: No.

Okay

L392: What was the size of the area analyzed? What was the mode of analysis (presumably scanning ion imaging)? If scanning, what was the pixel number and dwell time?

Response: 80x80 μm imaged area, scanning ion imaging, pixels 256x 256. Dwell times: 12C14N = 1 sec., 12C15N = 5 sec., 13C14N = 2 sec.; wt. time 0.8 sec, 100 cycles (l. 453-455).

Reviewer Response: Thank you for adding this information. As mentioned above; I am guessing it should be “milliseconds per pixel”.

L394: 6000 MRP is not sufficient to resolve 13C 14N- from 11B 16O-, which can interfere at mass 27 amu at relatively high abundance in many samples. The mass difference between 13C 14N- and 11B 16O- is ~2.2 milli-atomic mass units, which requires ~12,000 MRP to resolve, which the CAMECA ims-1280 can do. Boron is a common surface contaminant and it is present in seawater. The “background” mentioned for Fig. 3 E & F may in fact be BO- counts that have become significant relative to the 13C 14N- counts. Where the CN- count rates are high, BO- may not be significant. Potentially the authors can incorporate this potential source of variability into their statistical evaluation. I am concerned, however, that the number of control samples may limit their statistical power.

Response: Boron is a trace component in the silica frustule of diatoms (Mejía et al (2013) rather than a surface contaminant. As stated earlier our control samples (analysis in cells which have not been exposed to isotopic tracers) showed slightly higher 13C:12C than expected from the natural abundance of 13C:12C. However, this number was subtracted from the values recorded with amended samples when calculating excess 13C:12C. We have performed more studies on cultures of Skeletonema strains with 11,000 MRP and get similar values of C- and nitrate assimilation rates in the transition to the stationary phase as reported here.

Reviewer Response: In fact, boron is a common contaminant. The fact that it is in the frustule only makes the concern great, though I support the conclusion that the data are okay.

Authors response: The 15N:14N and 13C:12C can only be measured inside the cell after the frustule has been pre-sputtered away.

It would be instructive, but not necessary, to state why the CN isotopomers were used for measuring the 13C/12C ratio.

Response: They are brighter and have the advantage of allowing two isotope ratios from 3 peaks (l. 452-453).

Thank you for adding that.

Did the team also collect C2- for 13C/12C measurements? That requires lower MRP and could potentially have been collected simultaneously.

Response: No, this was not done.

Okay

L400-402: For the C and N assimilation rates, please cite one paper that has the exact procedure used or describe the procedure. Differences among the procedures cannot be reconciled by the reader.

Response: We have now cited Klawonn et al., 2016.

Okay

L403: Based on L166, cell size is determined by SIMS images. Was this done using the data processing software?

Response: No, cell sizes were determined in parallel samples by light microscopy as written in the Method section (l. 387-391). Winimage does not allow that as far as we are aware, so ROI's are placed visually.

Thank you for that clarification.

What were the criteria? 50% height of CN-?

Response: Each image of each cell was carefully examined by eye and the region of interest (ROI) was defined along the border of the $^{12}\text{C}^{14}\text{N}$ image and drawn by hand using the CAMECA software (l. 458-460). The border of the CN image was relatively easy define (Fig. 2)

Okay

Do the authors need to correct for erosion of the lateral dimensions of the cells during high current sputtering?

Response: We did not measure during the high-current sputtering, acquiring data only with the low current analytical beam. There was no substantial erosion based on the first and last image of the 100 cycles.

Reviewer Response: This is taken care of. I meant for the cell size, but now I understand that cell size was based parallel samples.

L427: The source of the derivation of this equation should be cited, and the limitations should be provided. Four papers are cited above in the text, which is not sufficiently specific. Note that this equation is inaccurate at high levels of enrichment relative to the pool because of its approximations. The authors should check that their data are inside the accurate range. Popa et al. 2007, ISME J, 1: 354 has a complete derivation of a related equation without approximations.

*Response: We are aware of the publication by Popa et al., which is a culture study in which transfer of fixed N_2 from heterocysts to vegetative cells is analyzed in *Anabaena* by use of nanosims. Only very few cells (< 10 cells) are analyzed in detail per time point. The very extensive analysis of isotopic values (using six equations) presented in that study may be appropriate with a low number cultured cells analyzed and with the aim of studying intercellular processes. This was not the aim of our study in which nutrient transfer between cells does not occur through intercellular transporters as in filamentous cyanobacteria.. The equation we have used is adapted from Montoya et al.(1996) and it has been used in a large number of SIMS publications (in contrast to the approach by Popa et al.) in the (nano) SIMS community . We consider that the equation we used is indeed valid and that values are within the accurate range for field populations within a plankton community study as also confirmed by our other quantitative studies of N_2 -fixation in the Baltic Sea where we found a 1:1 ratio between EA-IRMS and SIMS using similar enrichments and approaches because N_2 -fixation was confined to a well-defined group of filamentous cyanobacteria (Klawonn et al., 2016). We have used same approach in culture studies and find a very good agreement between growth rates calculated from SIMS measurements and those observed by cell counts over time (pers. unpubl. results).*

Reviewer Response: Thank you for this information. I am confused by why Montoya et al is not directly cited with the equation. My recollection of Adam et al is that it did not link Montoya et al to this equation, so if I am correct, I would recommend against using it and only citing one paper—presumably Klawonn et al., 2016 if it cites Montoya et al.—so that the reader does not have to look through three papers to find the full source.

Authors response: Montoya was cited in the original submission. Now we have included both.

Fig. 2: This image needs a scale bar.

Response: we have added this to the image

Okay.

Fig.3. Color scales are hard to read. What are the counts in A and B?

Response: These are counts per pixel. This information has been added to the figure legend.

Okay. This addresses my concern about possibly low total counts below.

They are relatively low for CN if that is a cumulative by pixel.

Response: They were sufficiently high for our purposes. As pointed out earlier follow-up experiments on C- and N-assimilation rates and their ratios in Skeletonema have been very reproducible.

Reviewer Response: Okay. The clarification in the previous response explains why the numbers can be low but still sufficient.

How are the ROIs defined?

Response: As mentioned above, the ROI were defined by eye using a freehand polygon tool in Winimage2 based on 12C14N images (l. 458-460).

Okay.

Fig. 4. I'm having a hard time with this figure. I am not seeing the relative changes indicated in the text. Perhaps these figures should be in the Supplemental, and instead the average and standard deviation of the data should be plotted on a single graph, along with the number of data points represented by each data point.

Response: It is important to document the large cell-to-cell variation in field populations – now included in the discussion (l. 236-251). The average and standard variations are plotted in Fig 5 and Fig. 6.

Reviewer Response: While I appreciate showing the variability, I do not think it helps to make the authors' case.

Author's response: The cell-to-cell variation in natural field populations is important to address. Furthermore, the figure shows how close the C- and N-assimilation ratio is to Redfield.

I find this graph confusing relative to the two experiments called out in the Methods.

Response: Fig. 4 shows the data from the first experiment, whereas Fig. 5 shows the data from both experiments.

Reviewer Response: It would be helpful to the reader to put this information in the captions.

Fig.5. It would be helpful to have statistical differences indicated in the graph. Also, it would be helpful to have the number of analyses represented in each bar shown in parentheses to aid reader interpretation.

Response: the requested information is written in the text (line?). We believe that the figure gets overloaded and less clear when number of analyses shown in parentheses are added to the graphs. The number analyses are stated in the figure legend.

Reviewer Response: I suppose this is an editorial decision. Busy or not, I think this presentation undermines the authors' case, particularly in Figure 6, where it doesn't look as if there are significant differences and it is extra work to try to figure out which comparisons are statistically significant. At least using standard error of the mean would help, or perhaps a non-parametric variance if for small sample numbers.

Note omission of "(D)" in Figure 5 caption, L771.

Fig.6. "deviation" is misspelled. As above, indicating statistical difference and number of analyses for each bar would be helpful.

Response: We have corrected the spelling accordingly. The requested information was and is written in the text (line?). We believe that the figure gets "too busy" when every information is added to the graphs.

See above.

Authors response: We have kept information about significance to the text, because we do multiple comparisons.

REVIEWERS' COMMENTS:

Reviewer #4 (Remarks to the Author):

I am a statistician and naturally focused on the statistical analysis presented in this paper. About this, I have three general comments. First, beginning on line 98, the Authors present three hypotheses that they wish to test. The reader might reasonably expect that the paper would include three statistical tests (and related analyses). In fact, there are many more than that, and the relation of each to the main hypotheses is not always clear. The Authors should present a brief roadmap along the lines of: here is an hypothesis, here is how we propose to test it, etc. Analyses with no substantial bearing on the main hypotheses should be omitted. Second, the Authors make very frequent use of two-way analysis of variance (ANOVA) without being sufficiently explicit about the hypothesis being tested (in general, it is almost always possible to infer this from the text) and the two factors in this analysis. For example, I cannot tell what these factors are in the very first application beginning on line 121. The Authors should preface each test with a statement like: Here, we use 2-way ANOVA to test for (fill in the blank) with factors (fill in the blank). Or something. Third, the Authors commonly present point estimates without any measure of uncertainty, like a standard error.

Response to reviewer:

Reviewer #4 (Remarks to the Author):

I am a statistician and naturally focused on the statistical analysis presented in this paper. About this, I have three general comments. First, beginning on line 98, the Authors present three hypotheses that they wish to test. The reader might reasonably expect that the paper would include three statistical tests (and related analyses). In fact, there are many more than that. and the relation of each to the main hypotheses is not always clear. The Authors should present a brief roadmap along the lines of: here is an hypothesis, here is how we propose to test it, etc. Analyses with no substantial bearing on the main hypotheses should be omitted. Second, the Authors make very frequent use of two-way analysis of variance (ANOVA) without being sufficiently explicit about the hypothesis being tested (in general, it is almost always possible to infer this from the text) and the two factors in this analysis. For example, I cannot tell what these factors are in the very first application beginning on line 121. The Authors should preface each test with a statement like: Here, we use 2-way ANOVA to test for (fill in the blank) with factors (fill in the blank). Or something.

Author's response: In the revised manuscript, we have followed the suggestions and explained our statistics with a better reference to our hypotheses and omitted statistics which are not essential for the outcome of the study.

Reviewer's comment: Third, the Authors commonly present point estimates without any measure of uncertainty, like a standard error.

Authors response: We have now explained in the figure legend of Fig. 3 that each data point represents rates at measured at single cell level (which cannot include s.d.). The error of the method is ca. 1%.